# Auto-regressive In-context Demonstration Selection

## Abstract

Effective demonstration selection is crucial for maximizing large language model (LLM) performance in few-shot in-context learning. Due to influences such as recency bias, the effectiveness of demonstrations depends heavily on their context relationship to the specific query, and on the ordering in which they are presented, making demonstration selection a complex combinatorial problem. To address these two challenges, we introduce AUTOSELECT, a novel framework that formulates demonstration selection as an auto-regressive sequential decision process. At each step, AUTOSELECT embeds the query and previously selected demonstrations into matrix representations to preserve structural information, and a trainable policy model sequentially selects the next best exemplar. To navigate the factorial space of demonstration permutations, our framework formulates a Kullback-Leibler (KL) regularized optimization problem, from which an optimal policy induces an optimal Plackett-Luce (PL) ranking over all possible demonstration sequences. Our theoretical analysis provides a principled learning objective: we prove that minimizing a tractable policy-level Cross-Entropy (CE) loss provably bounds the worst-case discrepancy between our policy's induced PL ranking and the optimal one, enabling tractable prioritization of high-quality sequences. Empirically, AUTOSELECT outperforms existing heuristic and learning-based methods across nine diverse datasets, achieving up to an 11% improvement over the strongest baseline. Our results are further supported by analytical studies and a case study, highlighting AUTOSELECT's key properties, as well as its transferability and generalizability.

## 1 Introduction

Large language models (LLMs) have demonstrated remarkable capabilities across diverse applications, including complex reasoning and code generation (Azerbayev et al., 2024; Imani et al., 2023; Shao et al., 2024). A key driver of inference-time performance is *few-shot in-context learning* (ICL), which enables LLMs to adapt to new tasks during inference using only a small number of demonstrations (exemplars) in the prompt (Brown et al., 2020; Achiam et al., 2023; Anthropic, 2024), delivering strong performance while remaining computationally efficient at inference time.

Previous studies suggest that the effectiveness of few-shot ICL depends critically on two factors: the *content* of the selected demonstrations and their *ordering* (Lu et al., 2022; Zhao et al., 2021; Zhang et al., 2023). As illustrated in Fig. 1, influences such as recency bias can disproportionately affect model outputs and task performance (Peysakhovich & Lerer, 2023), making careful control over both aspects essential. The interaction between these factors gives rise to *compositional effects*, transforming *query-dependent* demonstration selection (Zhang et al., 2022; Chen et al., 2024b) into a complex combinatorial optimization problem: an exemplar highly effective for one query can be irrelevant, or even detrimental, for a different query; and the same set of exemplars can yield vastly different outcomes depending on their ordering (Dong et al., 2024). Many methods approach this combinatorial challenge by framing demonstration selection as a ranking problem, scoring candidate exemplars individually based on heuristic criteria such as semantic similarity (Reimers & Gurevych, 2019; Izacard et al., 2022). While they capture marginal content relevance, these methods can overlook pairwise and higher-order dependencies among exemplars and their ordering (the *compositional effects*) (Ye et al., 2023). This omission can lead to a critical bottleneck for optimizing inference-time ICL performance.

Figure 1: Illustration of the sensitivity of in-context learning to demonstration content and order. **Left**: An example with relevant *content* (speed calculation) aids performance, while an irrelevant one does not. **Right**: The same set of examples can yield different outcomes based on their *ordering*.

Given that an exhaustive search for the optimal ordered sequence remains computationally intractable, there is a clear need for a more holistic yet tractable selection process. To address this, we draw inspiration from the core mechanism of LLMs themselves: the **auto-regressive paradigm** (Radford et al., 2018; Brown et al., 2020). By framing demonstration selection as a sequential decision process, we construct high-quality demonstration sequences tractably. Rather than retrieving isolated examples, our approach *"composes"* an effective demonstration sequence one step at a time, conditioned on the query and prior selections, much like an LLM generates a sentence token by token.

*Proposed Framework.* Motivated by the query-dependent and ordering-sensitive nature, we introduce AUTOSELECT, a novel auto-regressive framework built on a cohesive design for few-shot in-context demonstration selection. This problem is particularly challenging: even a simpler scenario, namely scoring an ordering by summing (over all exemplar pairs) the advantage of showing one before the other for the query, will reduce to the NP-hard Linear Ordering Problem (Martí & Reinelt, 2011). Thus, instead of attempting a direct and intractable search in this factorial space, AUTOSELECT re-frames the task as a dynamic, sequential decision process that mirrors how LLMs generate text. Our approach constructs a demonstration sequence one step at a time, conditioned on the input query and previous selections. This formulation is powered by a trainable policy model that operates on 2D matrix embeddings of exemplars and queries, a synergistic design that preserves internal token structure and captures complex inter-exemplar relationships. Meanwhile, AUTOSELECT also incorporates an adaptive stopping mechanism, allowing the model to dynamically learn the optimal sequence length for various queries. This unifies the critical aspects of demonstration selection, including content and ordering, into a single tractable framework.

*Theoretical and Empirical Insights.* To navigate the vast combinatorial space, AUTOSELECT employs a principled and theoretically-grounded optimization procedure. We learn the demonstration selection policy by minimizing a tractable *policy-level Cross-Entropy (CE) loss*, guided by sequence-level rewards that inherently evaluate the *compositional effects* of both demonstration *content* and their *ordering*. This is not a heuristic choice, since our theoretical analysis provides an intuitive guarantee: by modeling the problem as learning a Plackett-Luce (PL) distribution over demonstration-sequence rankings, minimizing this CE objective provably minimizes an upper bound on the discrepancy towards the *optimal ranking*. This enables AUTOSELECT to efficiently prioritize high-quality demonstration sequences without exhaustive enumeration. On the other hand, AUTOSELECT's effectiveness is also validated through comprehensive empirical evaluation. Across nine diverse tasks, AUTOSELECT consistently outperforms heuristic and learning-based baselines, achieving up to an 11% improvement over the strongest baseline. Notably, AUTOSELECT can deliver superior performance with considerably *fewer* demonstrations than top-$k$ retrieval, underscoring the practical value of optimizing demonstration quality rather than quantity alone. We further support these findings with extensive analytical studies on AUTOSELECT's behaviors, as well as demonstrate its generalization capabilities via a cross-task transferability case study.

## 2 RELATED WORKS

**Auto-regressive Paradigm.** The auto-regressive paradigm is foundational to sequential modeling and significantly advanced by the Transformer architecture (Vaswani et al., 2017). This formulation intrinsically supports the success of LLMs (Achiam et al., 2023; Anthropic, 2024; Grattafiori et al., 2024; Shao et al., 2024; Guo et al., 2025), and its effectiveness also extends to other modalities such as image and video generation (Weng et al., 2024; Tian et al., 2024). Crucially, recent adaptations of the auto-regressive paradigm to sequential decision-making, modeling reinforcement learning (RL) trajectories as sequences (Chen et al., 2021; Zheng et al., 2022; Lee et al., 2022), demonstrates its

capability for complex sequential, ordered selection tasks. Thus, we formulate in-context demonstration selection as an auto-regressive problem and propose AUTOSELECT, which effectively accounts for both the input query and the crucial effect of exemplar ordering on demonstration selection.

**Few-shot In-context Demonstration Selection.** The performance of few-shot in-context learning is highly dependent on the chosen exemplars, including their quantity (Li et al., 2023), formatting (Jiang et al., 2020), and ordering (Zhao et al., 2021; Lu et al., 2022; Dong et al., 2024). A dominant paradigm for this task is *retrieval-based selection*, which generally treats the problem as a top-$k$ ranking task (Margatina et al., 2023). These methods include sparse retrieval methods such as BM25 (Robertson et al., 2009) and dense retrieval methods with learned semantic embeddings like Contriever (Izacard et al., 2022). There are also retrieval methods adopting the "select-then-rank" framework based on similarity and model-dependent scores (Peng et al., 2024). Meanwhile, *learning-based* approaches have been developed to capture richer inter-exemplar relationships. Some formulate the task as subset selection, using contrastive learning and Determinantal Point Processes to encourage diversity (Ye et al., 2023; Rubin et al., 2022). Others model the selection as a Markov Decision Process, training a policy with exemplar-level rewards (Zhang et al., 2022; Chen et al., 2024b). For example, Wang et al. (2025) formulate this as an RL problem to jointly optimize for both task-relevance and exemplar diversity. Other research optimizes a single and static exemplar sequence shared across all queries for a given task (Min et al., 2022; Wu et al., 2024), where this task-level selection can help reduce inference-time overhead (Purohit et al., 2024). One category of these works formulates this as a subset selection problem to identify high-performing exemplars (Wu et al., 2024; Purohit et al., 2024; 2025), which can either consider exemplar ordering or remain order-independent. There are also static approaches that apply an iterative construction to build a single set that optimizes for group fairness alongside accuracy (Halim et al., 2025). Meanwhile, Purohit et al. (2025) first select a static pool of candidate subsets at the task level and then choose from this reduced set at inference time. In contrast, AUTOSELECT constructs the demonstration sequence one step at a time, conditioning each choice on the full context of the query and previously selected exemplars. This enables AUTOSELECT to explicitly model crucial ordering and compositional effects using only sequence-level supervision, rather than potentially expensive exemplar-level supervision.

## 3 PRELIMINARIES AND PROBLEM DEFINITION

**Auto-regressive Paradigm.** Given a query $\boldsymbol{x}$, an auto-regressive model (e.g., a language model) sequentially generates each output element, by sampling from the conditional probability $\mathcal{P}(\boldsymbol{e}_{i_t} \mid \boldsymbol{x}, \tau_{<t})$, where $\tau_{<t} = (\boldsymbol{e}_{i_1}, \ldots, \boldsymbol{e}_{i_{t-1}})$ denotes the elements chosen before position $t$. Then, the $t$-th element $\boldsymbol{e}_{i_t}$ is generated (e.g., sampled from the vocabulary) subsequently, which will terminate upon stopping criteria (e.g., reaching a maximum sequence length). This process naturally aligns with modern LLM generation (Li & Liang, 2021), and subsequently motivates our *"auto-regressive in-context demonstration selection"* problem below.

**Auto-regressive In-context Demonstration Selection.** Suppose we have $N$ candidate demonstration examples (exemplars), represented by $\mathcal{E} := \{\boldsymbol{e}_1, \ldots, \boldsymbol{e}_N\}$, where each exemplar $\boldsymbol{e}_i, i \in [N]$ refers to one query-answer pair. Here, given an input query $\boldsymbol{x}$, our policy model $\pi_\theta$, parameterized by $\theta$, needs to select an *ordered sequence* of unique exemplars, containing at most $T$ elements ($T \leq N$): $\tau = (\boldsymbol{e}_{i_1}, \boldsymbol{e}_{i_2}, \ldots, \boldsymbol{e}_{i_{|\tau|}})$, $|\tau| \leq T \leq N$. Our trainable policy model $\pi_\theta$ can be characterized by:

$$\pi_\theta(\tau \mid \boldsymbol{x}) = \prod_{t=1}^{|\tau|} \pi_\theta(\boldsymbol{e}_{i_t} \mid \boldsymbol{x}, \boldsymbol{e}_{i_1}, \ldots, \boldsymbol{e}_{i_{t-1}}), \tag{1}$$

where $\boldsymbol{e}_{i_t}$ refers to $t$-th element in the generated sequence (trajectory), and prefix $(\boldsymbol{x}, \boldsymbol{e}_{i_1}, \ldots, \boldsymbol{e}_{i_{t-1}})$ is fed into policy $\pi_\theta$ to choose the next exemplar, ensuring that our selections adapt to both the query and previously chosen exemplars. The number of chosen exemplars $|\tau|$ can differ for various queries $\boldsymbol{x}$. For the rest of the paper, we will use terms *"sequence"* and *"trajectory"* interchangeably.

Under the combined system of the policy $\pi_\theta$ and a fixed task-solving LLM, we denote $\mathcal{P}_\theta(\boldsymbol{y} \mid \tau, \boldsymbol{x})$ as the probability of the correct answer $\boldsymbol{y}$, given query $\boldsymbol{x}$ and exemplar sequence $\tau$. In this context, we aim to train parameters $\theta$ of policy model $\pi_\theta$, such that given an input query $\boldsymbol{x}$, the policy-generated demonstration sequence (trajectory) $\tau \sim \pi_\theta(\cdot|\boldsymbol{x})$ maximizes the likelihood of the correct answer:

$$\max_\theta \ \mathbb{E}_{(\boldsymbol{x},\boldsymbol{y})}\Big[\mathcal{P}_\theta(\boldsymbol{y} \mid \tau, \boldsymbol{x})\Big], \tag{2}$$

which encourages the policy $\pi_\theta$ to assign higher probabilities to exemplar sequences that guide the LLM towards correct answers, tailored to different input queries.

Figure 2: AUTOSELECT framework. We first embed candidate exemplars and the input query into matrix representations, then apply a trainable policy to sequentially select the next exemplar (or early stop) conditioned on the query and prior selections. After evaluating collected exemplar sequences, our proposed CE loss is minimized to align the policy's induced PL ranking with the optimal PL ranking, thereby prioritizing high-quality exemplar sequences.

## 4 PROPOSED FRAMEWORK: AUTOSELECT

**Framework Overview.** Fig. 2 illustrates the pipeline of AUTOSELECT framework. (1) [Subsec. 4.1] We first embed exemplars $\mathcal{E}$ and input queries $x$ into matrix embeddings to preserve their structural information. For the input query in each training episode, we collect trajectories (i.e., exemplar sequences) of varying lengths with our designed policy model, and validate corresponding trajectory-level rewards. (2) [Subsecs. 4.2 and 4.3] With the collected trajectories and their rewards, we train our policy $\pi_\theta$ by minimizing our proposed policy-level Cross-Entropy (CE) loss, thereby reducing the discrepancy between $\pi_\theta$ and the optimal policy $\pi^*$. This consequently refines the PL ranking induced by $\pi_\theta$, towards the optimal PL ranking induced by $\pi^*$, enabling $\pi_\theta$ to adaptively prioritize high-quality, query-specific exemplar sequences. Our pseudo-code is presented in Algs. 1 and 2.

### 4.1 TRAJECTORY GENERATION: POLICY MODEL AND TRAJECTORY ROLLOUTS

**(I) Matrix Embedding.** To preserve structural information (e.g., the token sequence ordering that enables context-aware understanding, and inter-token embedding relationship), we embed candidate exemplars $\mathcal{E}$ and the input query $x$ into individual embedding matrices, a strategy shown to effectively preserve structural information of text (Kim, 2014; Devlin et al., 2019; Khattab & Zaharia, 2020). Each embedding matrix is structured: *rows* correspond to tokens from the exemplars or query (padded to a fixed length); and *columns* represent the embedding vectors of these tokens, obtained through a pre-trained embedding (e.g., GPT-2 embedding for our experiments). We slightly abuse the notation, by also using $e \in \mathcal{E}$ to represent the exemplar *embedding matrix*. The input query $x$ will also be embedded into its *matrix representation* with this procedure.

**(II) Next-element Representation.** Recall that $e_{i_t}$ is $t$-th element in the exemplar sequence as in Eq. 1. Following the auto-regressive paradigm, given an input query $x$, denote the preceding $t-1$ chosen elements within the exemplar sequence as $\tau_{<t} = (e_{i_1}, e_{i_2}, \ldots, e_{i_{t-1}})$. We apply a trainable encoding model $\phi(\cdot)$, such as a Transformer-based architecture (Vaswani et al., 2017), to generate the *matrix representation* $z_t$ for the $t$-th selected exemplar $e_{i_t}$. This is achieved by processing the concatenated sequence $(x, e_{i_1}, \ldots, e_{i_{t-1}})$, and the resulting representation of the $t$-th element is

$$z_t := \phi(x \oplus e_{i_1} \oplus \cdots \oplus e_{i_{t-1}}) \tag{3}$$

where $\oplus$ denotes the concatenation operation. $z_t$ denotes the encoded matrix representation of the upcoming $t$-th element, which is shaped to match the padded token length and embedding dimension of exemplars and query matrices, enabling integration with matrix embeddings above.

**(III) Next-element Sampling.** We then select the $t$-th element by *sampling* from the distribution based on softmax-normalized distances (Mensink et al., 2013; Dong et al., 2015). These combined yield the probability distribution of our trainable policy $\pi_\theta$, for selecting the $t$-th element:

$$\pi_\theta(e \mid x, \tau_{<t}) := \frac{\exp\left(-\gamma \cdot \|z_t - e\|_F^2\right)}{\sum_{e \in (\mathcal{E} \setminus \tau_{<t}) \cup \{e_{[\text{EOS}]}\}} \exp\left(-\gamma \cdot \|z_t - e\|_F^2\right)}, \quad \forall e \in (\mathcal{E} \setminus \tau_{<t}) \cup \{e_{[\text{EOS}]}\}, \tag{4}$$

where candidate choices at each step include all remaining candidate exemplars, along with a special *End-of-Sequence (EOS) signal* $e_{[\text{EOS}]}$. This allows the model to dynamically determine the optimal length of the exemplar sequence. $\gamma > 0$ controls distribution skewness for next-element sampling. Policy model implementation for experiments is detailed in Appendix B.2.

**(IV) Exemplar Sequence Reward Evaluation.** Our policy $\pi_\theta$ is trained based on sequence-level rewards, which are derived from the downstream task performance on sampled reference query-answer pairs $(\boldsymbol{x}, \boldsymbol{y})$. Given a query $\boldsymbol{x}$ and label $\boldsymbol{y}$, for an exemplar sequence $\tau$, we define its reward as a direct empirical measurement of the sequence's effectiveness:

$$r(\boldsymbol{x}, \tau) := L\left(\boldsymbol{y},\ \text{LLM}(\boldsymbol{x}; \tau)\right). \tag{5}$$

Here, $\text{LLM}(\cdot;\ \cdot)$ is the LLM response and $L(\cdot,\ \cdot)$ is the evaluation metric (e.g., accuracy). This reward serves as the training signal for maximizing our primary objective (Eq. 2). Note that in practice, the reward is *efficiently computed* using only a tiny batch of sampled reference samples per training episode (details in Appendix B.2.3). Thus, our training phase is a modest, one-time offline investment that yields a policy enhancing performance with inference-time efficiency.

**(V) Collection of Full Trajectories & Sub-trajectories with Early Termination.** To enable effective policy training, we need an informative collection of exemplar sequences (trajectories) $\mathcal{T}$ for the input query $\boldsymbol{x}$. This is achieved through a specialized rollout procedure, as in Alg. 2:

1. For a query $\boldsymbol{x}$, we generate $K$ rollouts, each capped at length $T$. At each step, the policy $\pi_\theta$ samples either an exemplar from the remaining candidates, or the EOS signal $\boldsymbol{e}_{[\text{EOS}]}$.

2. When the EOS signal is selected, the current sub-trajectory is evaluated for its reward (Eq. 5) and stored (Algs. 2, line 8). To enable exploration of longer sequences with dependency, the rollout then continues by resampling a non-EOS exemplar.

3. The process concludes when the maximum length $T$ is reached, at which point the final *full trajectory* is also evaluated and stored into the collection $\mathcal{T}$ (Algs. 2, line 14).

This efficiently populates $\mathcal{T}$ with both (i) *early-terminated* sub-trajectories and (ii) *full-length* trajectories from the same rollouts, fostering *dependency* among them. It enables the policy to effectively learn not only *which exemplars to select*, but also *when to adaptively terminate the sequence*.

### 4.2 REINFORCEMENT LEARNING (RL) PROBLEM AND PL RANKING OF TRAJECTORIES

**Kullback-Leibler (KL)-regularized Reinforcement Learning (RL) Problem.** To enable policy optimization without the knowledge of the unknown optimal demonstration sequence, we adopt a standard KL-regularized RL objective below, as a practical surrogate for the learning objective in Eq. 2. Here, the commonly adopted KL-divergence term ensures stable policy optimization, by penalizing excessive deviation from a previous checkpoint $\pi_{\text{old}}$ (Schulman et al., 2017; Rafailov et al., 2024; Chen et al., 2024a). For an input query $\boldsymbol{x}$ and the corresponding trajectories generated, our objective is to train the policy model $\pi_\theta$ by solving:

$$\max_{\pi_\theta} \mathbb{E}_{\tau \sim \pi_\theta(\tau|\boldsymbol{x})}\left[r(\boldsymbol{x}, \tau)\right]\ -\ \beta \cdot \mathbb{D}_{\text{KL}}\left(\pi_\theta(\tau \mid \boldsymbol{x}) \,\|\, \pi_{\text{old}}(\tau \mid \boldsymbol{x})\right), \tag{6}$$

where $\tau = \{\boldsymbol{e}_{i_1}, \ldots, \boldsymbol{e}_{i_{|\tau|}}\}$ refers to the generated trajectory as defined in Eq. 1, namely the chosen exemplars with cardinality $|\tau|$. Reward evaluation $r(\cdot, \cdot)$ is formulated by Eq. 5, and coefficient $\beta > 0$ controls the regularization intensity. It has been shown that the above optimization problem leads to a closed-form solution (Peters & Schaal, 2007; Rafailov et al., 2024), such that the *optimal policy* $\pi^*$ solving Eq. 6 can be derived as

$$\pi^*(\tau|\boldsymbol{x}) = \frac{1}{Z(\boldsymbol{x})} \cdot \pi_{\text{old}}(\tau|\boldsymbol{x}) \exp\left(\beta^{-1} r(\boldsymbol{x}, \tau)\right), \tag{7}$$

where the partition function $Z(\boldsymbol{x}) = \sum_{\tau'} \pi_{\text{old}}(\tau'|\boldsymbol{x}) \exp\left(\beta^{-1} r(\boldsymbol{x}, \tau')\right)$ is intractable, as it is taken with respect to *all possible* trajectories. Intuitively, although we have access to rewards $r(\boldsymbol{x}, \tau)$ by Eq. 5, it is infeasible to directly apply the optimal policy $\pi^*$ to choose from all possible trajectories, as the trajectory volume will increase factorially along with number of candidate exemplars $\mathcal{E}$.

**PL Ranking of Trajectories.** After generating a trajectory collection $\mathcal{T}$ and evaluating their rewards, we need a principled way to *learn from their relative quality* and *prioritize high-quality exemplar sequences*. Thus, we formalize this using the PL model (Plackett, 1975; Luce et al., 1959), a standard probabilistic framework for modeling distributions over rankings based on utility scores. Given a trajectory collection $\mathcal{T} = \{\tau_i\}_{i=1}^{|\mathcal{T}|}$ for query $\boldsymbol{x}$, for a permutation $\sigma$ of trajectory indices $\{1, \ldots, |\mathcal{T}|\}$,

we have the optimal (reward-based) PL model induced by optimal policy $\pi^*$:

$$P_{\pi^*}(\sigma \mid \mathcal{T}, \boldsymbol{x}) := \prod_{i=1}^{|\mathcal{T}|} \frac{\exp\left(r(\boldsymbol{x}, \tau_{\sigma(i)})\right)}{\sum_{j=i}^{|\mathcal{T}|} \exp\left(r(\boldsymbol{x}, \tau_{\sigma(j)})\right)} = \prod_{i=1}^{|\mathcal{T}|} \frac{\exp\left(\beta \log \frac{\pi^*(\tau_{\sigma(i)}|\boldsymbol{x})}{\pi_{\text{old}}(\tau_{\sigma(i)}|\boldsymbol{x})}\right)}{\sum_{j=i}^{|\mathcal{T}|} \exp\left(\beta \log \frac{\pi^*(\tau_{\sigma(j)}|\boldsymbol{x})}{\pi_{\text{old}}(\tau_{\sigma(j)}|\boldsymbol{x})}\right)}. \quad (8)$$

where $\tau_{\sigma(i)}$ refers to the $i$-th ranked trajectory of permutation $\sigma$, and the second equality is by analogously transforming Eq. 7 to derive the closed-form representation of $r(\cdot, \cdot)$.

> **Why PL ranking?**
>
> Applying the PL model is directly motivated by our KL-regularized RL formulation (Eq. 6). The optimal policy $\pi^*$ (Eq. 7) assigns probabilities proportional to the exponential reward, $\pi^*(\tau \mid \boldsymbol{x}) \propto \exp(r(\boldsymbol{x}, \tau)/\beta)$, which matches the PL model's exponential scoring form. This makes the PL model naturally compatible with our goal of training $\pi_\theta$, so that its induced PL ranking matches the optimal PL ranking induced by $\pi^*$. Aligning with the optimal PL ranking trains $\pi_\theta$ to prioritize high-quality trajectories, thereby optimizing our main objective (Eq. 2).

**Bridging Policy and PL Ranking.** However, directly optimizing the discrepancy between PL models over permutations is infeasible, as there are $|\mathcal{T}|!$ possible permutations. To formulate a *tractable* policy training objective, we first theoretically bridge the PL ranking with policy optimization.

> **Proposition 4.1** (Equivalence of Optimal PL Ranking and Optimal Policy). *Consider a regularized RL problem in Eq. 6 and the associated reward function. A trainable policy $\pi_\theta$ is identical to the optimal policy $\pi^*$ (i.e., $\pi^* = \pi_\theta$) if and only if their PL ranking probabilities, $P_{\pi^*}(\sigma \mid \mathcal{T}, \boldsymbol{x}) = P_{\pi_\theta}(\sigma \mid \mathcal{T}, \boldsymbol{x})$, are equal for any possible trajectory collection $\mathcal{T}$.*

The proof of Proposition 4.1 is in Appendix E. This equivalence motivates our strategy of training $\pi_\theta$ to match the properties of the optimal policy $\pi^*$, with the ultimate goal of achieving the optimal PL ranking $P_{\pi^*}$. This relationship is also robust: by the invariance of the KL-regularized optimal policy to query-dependent reward shifts (Appendix F), the induced PL matching can faithfully capture the relative preferences among trajectories. Unfortunately, directly minimizing the universal discrepancy between $\pi^*$ and $\pi_\theta$ also remains infeasible: for a given input query $\boldsymbol{x}$, we only have access to a finite trajectory collection $\mathcal{T}$ rather than the full reward distribution. This motivates our practical policy-level Cross-Entropy (CE) loss, to be detailed in the next sub-section.

## 4.3 Practical Objective: Policy-level CE Loss for PL Discrepancy Minimization

**PL Discrepancy Minimization: Theoretical Intuition.** Here, since the partition function $Z(\boldsymbol{x})$ from Eq. 7 is intractable, we can compute the target probability distribution over trajectories induced by the optimal policy $\pi^*$ when restricted to a specific collection $\mathcal{T} = \{\tau_i\}_{i=1}^{|\mathcal{T}|}$. For any trajectory $\tau \in \mathcal{T}$, the probability conditioned on the collection $\mathcal{T}$ can be derived, by re-normalizing the expression (Eq. 7) over the trajectories from $\mathcal{T}$:

$$\pi^*(\tau|\mathcal{T}, \boldsymbol{x}) = \frac{\frac{1}{Z(\boldsymbol{x})} \cdot \pi_{\text{old}}(\tau|\boldsymbol{x}) \exp\left(\frac{r(\boldsymbol{x}, \tau)}{\beta}\right)}{\sum_{\tau' \in \mathcal{T}} \frac{1}{Z(\boldsymbol{x})} \cdot \pi_{\text{old}}(\tau'|\boldsymbol{x}) \exp\left(\frac{r(\boldsymbol{x}, \tau')}{\beta}\right)} = \frac{\pi_{\text{old}}(\tau|\boldsymbol{x}) \exp\left(\frac{r(\boldsymbol{x}, \tau)}{\beta}\right)}{\sum_{\tau' \in \mathcal{T}} \pi_{\text{old}}(\tau'|\boldsymbol{x}) \exp\left(\frac{r(\boldsymbol{x}, \tau')}{\beta}\right)}, \quad (9)$$

which provides the target relative likelihoods among the trajectories in the collection $\mathcal{T}$, with regard to optimal policy $\pi^*$ (Eq. 7). Analogously, we can define *conditional probability distribution* for our learnable policy $\pi_\theta(\tau|\mathcal{T}, \boldsymbol{x}) = \frac{\pi_\theta(\tau|\boldsymbol{x})}{\sum_{\tau' \in \mathcal{T}} \pi_\theta(\tau'|\boldsymbol{x})}$. With the above preliminaries, we motivate our training objective with Theorem 4.2 below, which shows that over a finite trajectory collection $\mathcal{T}$, the discrepancy between the PL rankings $P_{\pi^*}$ and $P_{\pi_\theta}$ can be minimized, by instead reducing the conditional discrepancy between the policies $\pi^*$ and $\pi_\theta$.

**Algorithm 1** AUTOSELECT (One Training Episode)

1: **Inputs:** $T$. $\gamma$. Number of trajectory rollouts $K$. Embedded $\mathcal{E}$ and $e_{\text{[EOS]}}$. Replay Buffer $\mathcal{B}$.
   ▷ **Generating New Trajectories with $\pi_\theta$**
2: $\pi_{\text{old}} \leftarrow \pi_\theta$. Trajectory Collection $\mathcal{T} \leftarrow \emptyset$.
3: Sample and embed query $\boldsymbol{x}$ as reference data.
4: **for** $k \in \{1, \ldots, K\}$ **do**
5:     $\mathcal{T}_k \leftarrow$ GenerateTrajectory$(\pi_\theta, \boldsymbol{x}, T, \gamma, k)$.
6:     $\mathcal{T} \leftarrow \mathcal{T} \cup \mathcal{T}_k$.
7: **end for**
   ▷ **Training with Instant Trajectories**
8: Compute CE loss $\mathcal{L}_{\text{CE}}$ (Eq. 11) with $\mathcal{T}$. Train $\pi_\theta$.
   ▷ **Training with Replay Buffer**
9: Sample small batch $\widehat{\mathcal{B}} \subseteq \mathcal{B}$ from replay buffer $\mathcal{B}$.
10: Calculate $\mathcal{L}_{\text{CE}}$ with $\widehat{\mathcal{B}}$ and update policy $\pi_\theta$.
11: Update replay buffer $\mathcal{B} \leftarrow \mathcal{B} \cup \{(\boldsymbol{x}, \mathcal{T})\}$.

**Algorithm 2** GenerateTrajectory $(\pi_\theta, \boldsymbol{x}, T, \gamma, k)$

1: Initialize trajectory $\tau \leftarrow ()$, collection $\mathcal{T}_k \leftarrow \emptyset$.
2: **for** $t = 1, \ldots, T$ **do**
3:     Get representation: $\boldsymbol{z}_t \leftarrow \phi(\boldsymbol{x} \oplus \tau_{<t})$.
4:     $p(\boldsymbol{e}) \leftarrow \text{Softmax}(-\gamma \|\boldsymbol{z}_t - \boldsymbol{e}\|_F^2)$,
                        $\forall \boldsymbol{e} \in (\mathcal{E} \setminus \tau_{<t}) \cup \{\boldsymbol{e}_{\text{[EOS]}}\}$.
5:     Sample $\boldsymbol{e}_{i_t} \sim \text{Categorical}(p(\boldsymbol{e}))$.
6:     **if** $\boldsymbol{e}_{i_t} == \boldsymbol{e}_{\text{[EOS]}}$ **then**
7:         Obtain reward for current $\tau$ (Eq. 5) .
8:         Update $\mathcal{T}_k \leftarrow \mathcal{T}_k \cup \{(\tau_{<t}, \boldsymbol{e}_{\text{[EOS]}})\}$.
9:         $p'(\boldsymbol{e}) \leftarrow \text{Softmax}(-\gamma \|\boldsymbol{z}_t - \boldsymbol{e}\|_F^2)$,
                            $\forall \boldsymbol{e} \in (\mathcal{E} \setminus \tau_{<t})$.
10:        Re-sample: $\boldsymbol{e}_{i_t} \sim \text{Categorical}(p'(\boldsymbol{e}))$.
11:    **end if**
12:    Update trajectory $\tau \leftarrow (\tau_{<t}, \boldsymbol{e}_{i_t})$.
13: **end for**
14: Obtain reward and $\mathcal{T}_k \leftarrow \mathcal{T}_k \cup \{(\tau_{\leq T}, \boldsymbol{e}_{\text{[EOS]}})\}$.
15: **return** Trajectory collection $\mathcal{T}_k$

---

**Theorem 4.2** (PL Ranking Optimization via CE Loss Minimization (Informal)). *Given input query $\boldsymbol{x}$ and trajectory collection $\mathcal{T}$, let $\sigma$ be any permutation of trajectories in $\mathcal{T}$. The maximum absolute difference, between the probabilities assigned to $\sigma$ by the PL models of policies $\pi^*$ and $\pi_\theta$, can be bounded as*

$$\max_\sigma \left| P_{\pi^*}(\sigma \mid \mathcal{T}, \boldsymbol{x}) - P_{\pi_\theta}(\sigma \mid \mathcal{T}, \boldsymbol{x}) \right| \leq \Phi\big(\mathcal{L}_{CE}^{\mathcal{T}}(\pi^*, \pi_\theta)\big), \quad (10)$$

*where $\mathcal{L}_{CE}^{\mathcal{T}}(\pi^*, \pi_\theta)$ is the CE loss conditioned on $\mathcal{T}$, between $\pi^*(\tau|\mathcal{T}, \boldsymbol{x})$ and $\pi_\theta(\tau|\mathcal{T}, \boldsymbol{x})$. $\Phi(\mathcal{L})$ decreases with $\mathcal{L}$, and $\Phi(\mathcal{L}) = 0$ when $\mathcal{L}_{CE}^{\mathcal{T}}(\pi^*, \pi_\theta)$ reaches its minimum, i.e., $\mathcal{L}_{CE}^{\mathcal{T}}(\pi^*, \pi^*)$.*

The formal theorem and proof are provided in Appendix D. Theorem 4.2 suggests that the maximum PL ranking discrepancy of any permutation can be upper bounded by a decreasing function of the CE loss. Consequently, this indicates that minimizing the CE loss conditioned on the trajectory collection $\mathcal{T}$ can serve as a feasible training objective, for learning towards the optimal PL ranking.

**Practical Training Objective: Minimizing Policy Discrepancy with CE Loss.** Motivated by above insights, to train our policy $\pi_\theta$ to match the target distribution from the optimal policy $\pi^*$, we first denote the training data $\mathcal{D}$: a batch of query-trajectory-collection pairs $(\boldsymbol{x}, \mathcal{T})$ with corresponding rewards. For each pair, we can treat the target distribution $\pi^*(\tau|\mathcal{T}, \boldsymbol{x})$ from Eq. 9 as "soft labels" over the trajectories $\tau \in \mathcal{T}$. Then, we propose to minimize the CE loss, between this target distribution and the distribution predicted by our learnable policy $\pi_\theta(\tau|\mathcal{T}, \boldsymbol{x})$, defined as:

$$\mathcal{L}_{\text{CE}}(\mathcal{D}) := -\frac{1}{|\mathcal{D}|} \sum_{(\boldsymbol{x}, \mathcal{T}) \in \mathcal{D}} \sum_{\tau \in \mathcal{T}} \Big[ \pi^*(\tau|\mathcal{T}, \boldsymbol{x}) \cdot \log \pi_\theta(\tau|\mathcal{T}, \boldsymbol{x}) \Big]$$

$$= -\frac{1}{|\mathcal{D}|} \sum_{(\boldsymbol{x}, \mathcal{T}) \in \mathcal{D}} \sum_{\tau \in \mathcal{T}} \left[ \frac{\pi_{\text{old}}(\tau|\boldsymbol{x}) \exp\left(\frac{r(\boldsymbol{x}, \tau)}{\beta}\right)}{\sum_{\tau' \in \mathcal{T}} \pi_{\text{old}}(\tau'|\boldsymbol{x}) \exp\left(\frac{r(\boldsymbol{x}, \tau')}{\beta}\right)} \log\left(\frac{\pi_\theta(\tau|\boldsymbol{x})}{\sum_{\tau' \in \mathcal{T}} \pi_\theta(\tau'|\boldsymbol{x})}\right) \right]. \quad (11)$$

Minimizing $\mathcal{L}_{\text{CE}}$ helps align with the optimal PL ranking, thereby prioritizing high-quality trajectories.

**Training with Instant Trajectories & Replay Buffer.** We apply *multi-episode* training for $\pi_\theta$. In each training episode, we receive an input query $\boldsymbol{x}$ and will generate a trajectory collection $\mathcal{T}$, along with corresponding rewards evaluated. (1) *Instant Trajectory Update*: Update $\pi_\theta$ (Alg. 1, line 8) by minimizing the CE loss (Eq. 11) computed on the current episode's collected trajectories $\mathcal{T}$ and their rewards. (2) *Replay-buffer Update*: Sample a small batch of past (query, trajectory-collection) pairs (Alg. 1, lines 9-11) and further update $\pi_\theta$ using the CE loss on this batch.

**Inference-time Demonstration Selection.** During the inference time, the learned $\pi_\theta$ will generate exemplar sequences for testing queries, following *Steps (I) to (III) in Subsec. 4.1*. For each query, its demonstration selection terminates upon selecting $\boldsymbol{e}_{\text{[EOS]}}$ or reaching maximum length $T$.

Table 1: Comparison of AUTOSELECT with seven baselines (mean performance $\pm$ standard deviation over 3 seeds) with average ranks. Best results are shown in **bold** with dark green shading, and second-best are underlined with light blue shading. The final column reports AUTOSELECT's improvement percentage over the second-best method (values in parentheses exclude greedy-oracle).

| Task \ Method | Learning-free | | | Oracle | Learning-based | | | Ours | |
|---|---|---|---|---|---|---|---|---|---|
| | random | max-entropy | re-ordering | greedy-oracle | ActRL | CEIL | EASE | AUTOSELECT | Impro. (w/o oracle) |
| AGNews | $0.767_{\pm 0.027}$ | $0.774_{\pm 0.035}$ | $0.773_{\pm 0.040}$ | $\mathbf{0.848_{\pm 0.015}}$ | $0.819_{\pm 0.036}$ | $0.812_{\pm 0.011}$ | $0.826_{\pm 0.022}$ | $0.845_{\pm 0.005}$ | -0.4% (+2.3%) |
| Amazon | $0.911_{\pm 0.008}$ | $0.938_{\pm 0.000}$ | $0.939_{\pm 0.007}$ | $\underline{0.943_{\pm 0.006}}$ | $0.922_{\pm 0.004}$ | $0.925_{\pm 0.010}$ | $0.924_{\pm 0.003}$ | $\mathbf{0.951_{\pm 0.004}}$ | +0.8% (+1.3%) |
| SST-2 | $0.900_{\pm 0.014}$ | $0.903_{\pm 0.027}$ | $0.908_{\pm 0.016}$ | $\underline{0.934_{\pm 0.001}}$ | $0.916_{\pm 0.021}$ | $0.912_{\pm 0.061}$ | $0.922_{\pm 0.016}$ | $\mathbf{0.946_{\pm 0.003}}$ | +1.3% (+2.6%) |
| Trec | $0.217_{\pm 0.025}$ | $0.277_{\pm 0.009}$ | $0.303_{\pm 0.049}$ | $0.370_{\pm 0.018}$ | $0.283_{\pm 0.040}$ | $\underline{0.375_{\pm 0.046}}$ | $0.373_{\pm 0.055}$ | $\mathbf{0.393_{\pm 0.023}}$ | +4.8% (+4.8%) |
| Winowhy | $0.454_{\pm 0.030}$ | $0.443_{\pm 0.033}$ | $0.487_{\pm 0.070}$ | $0.589_{\pm 0.070}$ | $0.478_{\pm 0.050}$ | $\underline{0.591_{\pm 0.037}}$ | $0.580_{\pm 0.005}$ | $\mathbf{0.657_{\pm 0.012}}$ | +11.2% (+11.2%) |
| Epi._reasoning | $0.463_{\pm 0.012}$ | $0.461_{\pm 0.029}$ | $0.470_{\pm 0.007}$ | $\underline{0.561_{\pm 0.021}}$ | $0.482_{\pm 0.039}$ | $0.546_{\pm 0.043}$ | $0.532_{\pm 0.012}$ | $\mathbf{0.601_{\pm 0.012}}$ | +7.1% (+10.1%) |
| Timedial | $0.654_{\pm 0.066}$ | $0.620_{\pm 0.033}$ | $0.683_{\pm 0.042}$ | $0.712_{\pm 0.039}$ | $0.709_{\pm 0.029}$ | $0.712_{\pm 0.014}$ | $\underline{0.715_{\pm 0.011}}$ | $\mathbf{0.738_{\pm 0.008}}$ | +3.2% (+3.2%) |
| Hyperbaton | $0.516_{\pm 0.037}$ | $0.508_{\pm 0.026}$ | $0.516_{\pm 0.015}$ | $0.551_{\pm 0.026}$ | $0.573_{\pm 0.041}$ | $\underline{0.610_{\pm 0.021}}$ | $0.592_{\pm 0.047}$ | $\mathbf{0.663_{\pm 0.011}}$ | +8.7% (+8.7%) |
| AQuA | $0.348_{\pm 0.014}$ | $0.346_{\pm 0.024}$ | $0.355_{\pm 0.008}$ | $\underline{0.374_{\pm 0.016}}$ | $0.349_{\pm 0.010}$ | $0.344_{\pm 0.013}$ | $0.332_{\pm 0.011}$ | $\mathbf{0.395_{\pm 0.002}}$ | +5.6% (+11.3%) |
| Avg. Rank | 7.1 | 6.9 | 5.2 | $\underline{2.7}$ | 5.0 | 3.8 | 4.0 | **1.1** | \ |

# 5 EXPERIMENTS

**Experiment Settings.** We involve nine datasets with diverse specifications, including four commonly evaluated datasets (AGNews, Amazon, SST-2, Trec) in existing demonstration selection works (Zhao et al., 2021; Zhang et al., 2022; Li et al., 2023), four BigBench (bench authors, 2023) tasks (Winowhy, Epistemic_reasoning, Timedial, Hyperbaton) for testing LLM's few-shot induction and reasoning capabilities, and math reasoning dataset AQuA (Ling et al., 2017). Analogous to previous works on few-shot demonstration selection (Zhang et al., 2022; Wu et al., 2024), we set maximum sequence length to 4. For baselines, we involve (1) heuristic learning-free methods: random, max-entropy, re-ordering; (2) oracle-based method: greedy-oracle (Zhang et al., 2022) that selects the best candidate at each position via exhaustive enumeration, which is significantly more costly than other baselines and AUTOSELECT; (3) and three learning-based methods: Active Example Selection by RL (ActRL) (Zhang et al., 2022), CEIL (Ye et al., 2023), EASE (Wu et al., 2024). Qwen2.5-3B (Yang et al., 2025) is applied as our task-solving LLM. Detailed descriptions are in Appendix B.

We first present main empirical results: few-shot in-context learning experiments and the discussion of AUTOSELECT properties (Subsec. 5.1), followed by complementary comparisons with retrieval-based baselines under various settings (Subsec. 5.1.1). We then present a case study demonstrating AUTOSELECT's transferability and generalizability, under both direct-transfer and adaptation settings (Subsec. 5.2). In addition, we also provide complementary experiments (e.g., results across different LLM families, hyper-parameter study, and efficiency, inference-time analysis) in Appendix C.

## 5.1 FEW-SHOT IN-CONTEXT LEARNING WITH DEMONSTRATION SELECTION

**Main Results.** In Table 1, AUTOSELECT can generally outperform strong baselines, benefiting from its effective policy design and the auto-regressive paradigm. The consistent outperformance of the re-ordering method against the random baseline *empirically validates the importance of exemplar ordering*, a critical factor our AUTOSELECT is designed to exploit. While AUTOSELECT's improvement is marginal for saturated and less difficult tasks such as AGNews, AUTOSELECT can achieve substantial improvements on challenging ones, including Trec, four reasoning tasks, and math dataset AQuA. CEIL can generally outperforms EASE, particularly on challenging reasoning tasks, highlighting the importance of query-aware selection over fixed exemplars. While greedy-oracle achieves strong performance on certain tasks, it needs to exhaustively enu-

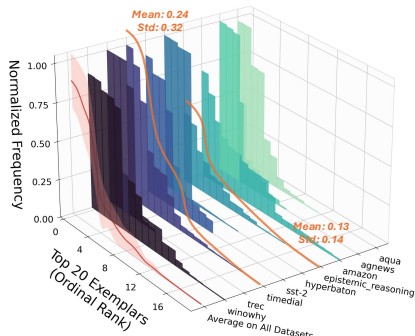

Figure 3: Top-20 exemplar selection frequencies across tasks.

merate all exemplars and all the corresponding rewards, making it significantly more computationally expensive than AUTOSELECT and other baselines. But, greedy-oracle still overlooks exemplar compositional effects, leading to sub-optimal performance.

**Properties.** From Fig. 3, AUTOSELECT can adaptively apply different selection strategies across tasks, while using $\sim 3$ exemplars on average (Fig. 10) with EOS mechanism. This highlights its ability to capture task-dependent exemplar utility. AUTOSELECT also demonstrates strong performance across LLM families and scales (Appendix C.1), and yields consistent gains for increasing maximum

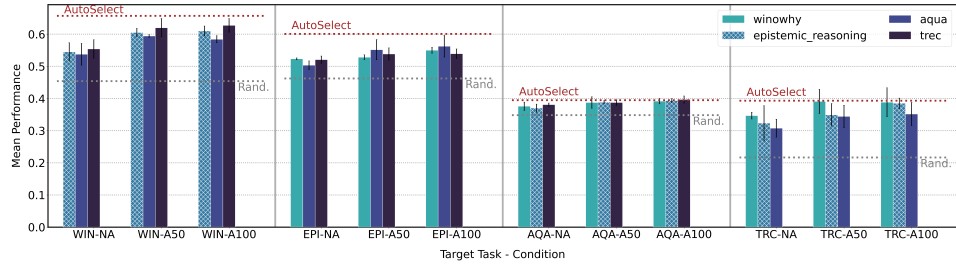

Figure 5: Transferability and generalization of trained policies from source tasks (legend) to target tasks (WIN: Winowhy; EPI: Epistemic_reasoning; AQA: AQuA; TRC: Trec) with Qwen-2.5-3B. Horizontal lines "Rand." (Random) and "AutoSelect" are reference performance from Table 1. X-axis: Target Task - Condition (NA: No Adaptation; A50/A100: 50/100 Adaptation Episodes).

sequence lengths up to $T = 16$ (Appendix C.2). Regarding efficiency, AUTOSELECT strikes a strong balance between *computational cost* and performance (Appendix C.3), leveraging one-time offline policy training to enable efficient and effective inference-time demonstration selection.

### 5.1.1 COMPLEMENTARY COMPARISONS WITH RETRIEVAL-BASED METHODS.

Table 2: Comparison with Retrieval-based Baselines.

| Method \ Task | Winowhy | Epi._reasoning | AQuA | Trec |
|---|---|---|---|---|
| Random | $0.454 \pm 0.030$ | $0.463 \pm 0.012$ | $0.348 \pm 0.014$ | $0.217 \pm 0.025$ |
| BM25 | $0.519 \pm 0.011$ | $0.497 \pm 0.003$ | $0.359 \pm 0.002$ | $0.364 \pm 0.004$ |
| Contriever | $0.538 \pm 0.027$ | $0.504 \pm 0.014$ | $0.357 \pm 0.005$ | $0.361 \pm 0.014$ |
| top-$k$ (Qwen2.5-3B Emb.) | $0.524 \pm 0.014$ | $0.501 \pm 0.013$ | $0.359 \pm 0.003$ | $0.375 \pm 0.014$ |
| CEIL | $0.591 \pm 0.037$ | $0.546 \pm 0.043$ | $0.344 \pm 0.012$ | $0.375 \pm 0.046$ |
| AUTOSELECT | $\mathbf{0.657 \pm 0.012}$ | $\mathbf{0.601 \pm 0.012}$ | $\mathbf{0.395 \pm 0.002}$ | $\mathbf{0.393 \pm 0.023}$ |

Despite learning-based CEIL, we also compare against three retrieval-based methods with pretrained embeddings: BM25 (Robertson et al., 2009), Contriever (Izacard et al., 2022), and a top-$k$ method (Margatina et al., 2023) with native Qwen2.5-3B embeddings.

From Table 2, AUTOSELECT can generally outperform these three baselines, underscoring the value of modeling exemplar interactions instead of relying on exemplar-level similarity or ranking scores alone. This indicates that exemplar selection guided by an auto-regressive policy can more effectively identify informative and task-relevant demonstrations, outperforming static heuristics and fixed similarity measures. AUTOSELECT performs particularly well on challenging reasoning datasets such as "Winowhy" and "Epistemic_reasoning", by jointly capturing query content and exemplar dependencies to guide demonstration selection more effectively.

**Top-$k$ w/ Enhanced Knowledge.** We further compare with a top-$k$ variant (Margatina et al., 2023) with enhanced knowledge and Qwen2.5-3B embeddings: top-$k$-enhanced. Recall that reward are computed on a validation set (Appendix B.2.3) to promote generalizable policy training. For fair comparisons, baselines requiring supervision signals (e.g., greedy-oracle and learning-based) will similarly derive their supervision from the same validation set. In this context, top-$k$-enhanced will leverage and select exemplars from the union collection of the exemplar set $\mathcal{E}$ and the validation set. In Fig. 4, AUTOSELECT can achieve stronger performance with considerably fewer demonstrations than top-$k$-enhanced, with advantages on larger $T$. This

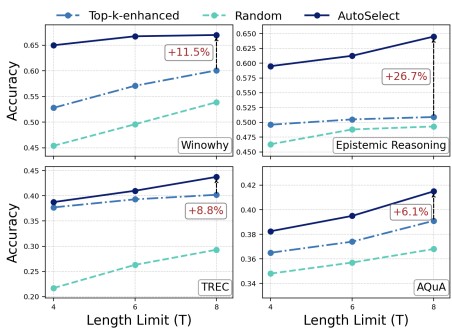

Figure 4: Comparison with an additional top-$k$ method w/ enhanced knowledge.

demonstrates that the learning capabilities are the key to the strong performance of AUTOSELECT.

### 5.2 CASE STUDY: TRANSFERABILITY AND GENERALIZABILITY OF TRAINED POLICIES

We also include a case study on the transferability and generalizability of trained policy $\pi_\theta$ across different tasks, under two scenarios: (1) direct transfer to a new task without further training, and (2) transfer with a small number of adaptation episodes on the target task.

**Results.** From Fig. 5, when directly transferring trained policy models onto different target tasks without adaptation, AUTOSELECT can already achieve better performance than simple heuristics. On the other hand, a simple adaptation of 50 or 100 episodes (e.g., around 13 minutes on the "Trec" task for 50 episodes) can further improve its transferability. Notably, for "AQuA" and "Trec", the adapted policy can achieve performance comparable to task-specific optimization results (Table 1). Policies trained on "Winowhy" and "Epistemic_reasoning" tend to demonstrate strong generalization,

as these tasks enable the policy to learn generalizable reasoning and justification patterns. These results demonstrate AUTOSELECT's strong transferability capability and potential, suggesting future extensions such as multi-task generalization, which we plan to explore in future works.

## 6 CONCLUSION

We formulate the problem of auto-regressive in-context demonstration selection and introduce a novel framework, AUTOSELECT, to solve this problem. Utilizing a trained policy model, AUTOSELECT can effectively perform query-specific and ordering-aware exemplar selection for LLM few-shot in-context learning during inference. Our theoretically grounded optimization procedure, with the proposed policy-level Cross-Entropy loss, learns toward the optimal PL ranking from sequence-level rewards, efficiently bypassing exhaustive enumeration. AUTOSELECT empirically outperforms strong baselines across nine datasets, while demonstrating robust generalization and adaptive selection, which validates the effectiveness of the auto-regressive in-context demonstration selection paradigm and offers insights for future extensions, such as multi-task and cross-domain adaptation.

## ETHICS STATEMENT

The authors have read and are in full compliance with the ICLR Code of Ethics. This paper introduces AUTOSELECT, an auto-regressive framework that advances LLM in-context learning by automating query-specific, ordered demonstration selection. This enhances LLM utility, accessibility, and potential transparency through understandable exemplars, while reducing manual effort. Our research does not involve human subjects and utilizes only public benchmark datasets. While we do not foresee significant negative impacts directly from this foundational methodology, we acknowledge that any technology improving LLM capabilities is subject to potential downstream misuse. Moreover, the fairness of our framework is contingent on the provided exemplar data, and any inherent biases can, be reflected in the selections. We believe our work contributes positively to the development of more efficient and reliable language models.

## REPRODUCIBILITY STATEMENT

To ensure the reproducibility of our work, we provide detailed descriptions of our methodology and experiments. The core framework, AUTOSELECT, and the trajectory generation procedure are formally described in Algs. 1 and 2. Our theoretical claims, including the main theorem connecting the CE loss to the PL ranking discrepancy (Theorem 4.2) and the relationship between policy optimization and PL ranking (Proposition 4.1), are theoretically proven in Appendix D and E, respectively. A comprehensive account of our experimental setup is available in Section 5 and Appendix B. This includes detailed descriptions of all datasets and baselines (Appendix B.1), the instantiation of our trainable policy model (Appendix B.2), and a description of hyperparameters, optimizer settings, and architectural specifications (Appendix B.3). The datasets used are publicly available, and our source code is available in supplementary materials.

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

# Appendix

APPENDIX CONTENTS

## A  THE USE OF LARGE LANGUAGE MODELS (LLMS)

LLMs are integral to this work in two different capacities. Primarily, they serve as the *task-solvers*, for which our proposed AUTOSELECT framework selects in-context demonstrations. To evaluate the generalizability and compatibility of our method, we employed a diverse set of publicly available, pre-trained LLMs from various model families and scales, including models from the Qwen and LLaMA series, as detailed in our experiments (and Appendix B). In this primary role, the LLMs' parameters were kept frozen, as our work focuses on an inference-time optimization method. Secondly, in one of our analytical studies (Appendix C.4), a pre-trained language model (GPT-2 Medium) was adapted to function as an alternative backbone for the *policy model* itself, allowing for a comparative analysis against our proposed policy model architecture. LLMs are also occasionally used for editing, to improve paper presentation and sentence clarity.

## B  EXPERIMENT IMPLEMENTATION DETAILS AND COMPLEMENTARY DISCUSSIONS

### B.1  BASELINE AND DATASET DESCRIPTIONS.

Recall that we compare against seven baselines, categorized into three groups: (1) learning-free heuristic methods, (2) an oracle-based method, and (3) existing learning-based methods. They include:

- Type 1: *Heuristic Learning-free Baselines*
  - **Random**: This method chooses $T$ exemplars randomly as exemplar sequence.
  - **Max-entropy**: It requires access to the logits of task-solving LLM, and greedily selects examples that maximize classification entropy.
  - **Re-ordering** (Lu et al., 2022): To provide a controlled ablation on the *impact of sequence ordering*, this baseline operates on the exact same set of randomly sampled exemplars as the "random" baseline. Holding the content fixed, it then optimizes *only* the permutation (ordering) of these examples by selecting the order that maximizes classification entropy.

- Type 2: *Oracle-based Baseline*
  - **Greedy-oracle** (Zhang et al., 2022): At each step of exemplar selection, it greedily enumerates and evaluates all remaining candidates by appending each one to the current sequence and measuring its validation performance. For example, in a 5-shot scenario with a pool of 100 exemplars, once 4 have been chosen, it evaluates all 96 remaining candidates, requiring 96 separate validation runs. More generally, to validate exemplar sequences, greedy-oracle needs to query the task-solving LLM on the validation set for every possible combination of candidate exemplars at each selection step, which incurs a dramatically higher computational cost than other methods. This exhaustive enumeration is performed at every selection step: 100 runs for the first exemplar, 99 for the second, and so on through the fifth, making it significantly expensive in terms of computation.

- Type 3: *Existing Learning-based ICL Demonstration Selection Methods* [1]
  - **Active Example Selection by RL (ActRL)** (Zhang et al., 2022): It models the exemplar selection as a Markov Decision Process (MDP), and selects the exemplars with a Deep Q-network (DQN).
  - **CEIL** (Ye et al., 2023): It addresses in-context example selection by framing it as a subset selection task, employing Determinantal Point Processes (DPPs) to model the interplay between a given input and the in-context examples, with a contrastive learning objective.
  - **EASE** (Wu et al., 2024): It uses hidden embeddings from a pre-trained language model to represent ordered exemplar sequences and applies a neural bandit algorithm to optimize sequence formulation for each task, instead of query-aware exemplar selection.

We also provide dataset descriptions and exemplary query-answer pairs in Table 3.

---

[1] We omit empirical comparisons with an existing work (Chen et al., 2024b), due to the lack of publicly available official code implementation from the authors, and instead include the discussion in our Related Works section.

## B.2 Implementation Details: Exemplar Sequence Generation with Vision Transformer (ViT)-based Policy Model, EOS Signal Instantiation, and Reward Evaluation

In this subsection, we provide instantiation details for our policy model architecture (Appendix B.2.1), implementation details of the EOS signal (Appendix B.2.2), as well as the details of our reward evaluation (Appendix B.2.3).

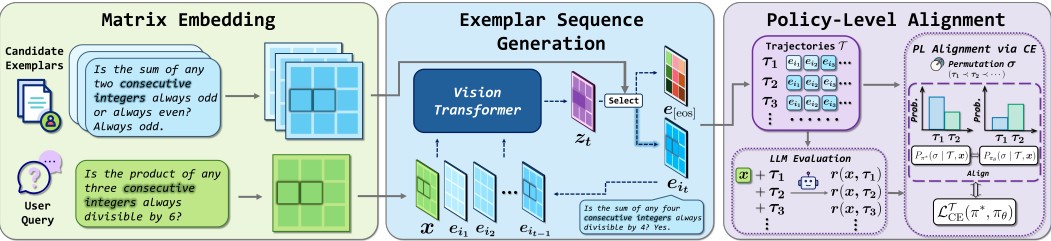

Figure 6: AUTOSELECT framework for auto-regressive demonstration selection. Queries and exemplars are first transformed into 2D matrix embeddings to preserve token-level structure. A ViT-based policy then auto-regressively processes the query and prior selections to generate a contextual representation, $z_t$, which guides the sequential selection of the next exemplar or an End-of-Sequence (EOS) signal. Finally, the generated sequences are evaluated by an LLM to obtain sequence-level rewards. These rewards are used to minimize a policy-level CE loss, aligning the policy's induced Plackett-Luce (PL) ranking with the optimal one to prioritize effective, query-aware sequences.

### B.2.1 ViT-based Policy Model

**Policy Model Architecture and Intuitions.** We instantiate our trainable encoding model $\phi(\cdot)$ and policy $\pi_\theta$ using a Vision Transformer (ViT) (Dosovitskiy et al., 2021), a natural choice for our auto-regressive selection process over 2D matrix embeddings. The pipeline instantiation is illustrated in Fig. 6. Recall that unlike methods requiring flattened vector inputs, our matrix representation preserves the vital token-level sequential structure of each exemplar. The ViT is uniquely suited to process our 2D structured matrix representations, allowing it to capture both internal token relationships and inter-exemplar dependencies.

At each step $t$, the ViT processes the sequence of matrix embeddings for the query $x$ and preceding exemplars $(e_{i_1}, \ldots, e_{i_{t-1}})$ to synthesize the context into a output matrix representation $z_t$. This matrix acts as a dynamic prototype to guide the selection of the next exemplar, making this synergistic design a powerful and well-justified choice for our framework. Our implementation choice for the encoding model $\phi(\cdot)$ is also validated by an ablation study and visualization results in Appendix C.4. Our ViT-based encoding model can generally achieve better performance, compared with a GPT-2 encoding model that has significantly more trainable parameters than our ViT-based model.

**Next-element Representation.** Afterwards, at each step $t$ in the generation of an exemplar sequence, the policy needs to make a selection conditioned on the initial input query $x$ and all previously chosen exemplars, denoted by the prefix $\tau_{<t} = (e_{i_1}, e_{i_2}, \ldots, e_{i_{t-1}})$. The core of our policy model is a trainable ViT that processes the sequence of matrix embeddings corresponding to this context, $(x, e_{i_1}, \ldots, e_{i_{t-1}})$, to produce a contextual representation for the current step.

The ViT outputs a sequence of matrix embeddings, one for each input element. We take the final matrix embedding from this output sequence as a summary representation, $z_t$, which encodes the entire preceding context. This representation effectively serves as a dynamic "query" for selecting the next exemplar. Formally, $z_t$ is obtained as:

$$z_t := \text{ViT}(x \oplus e_{i_1} \oplus \cdots \oplus e_{i_{t-1}})[-1, :, :] \tag{12}$$

where $\oplus$ denotes concatenation along the sequence dimension, and the indexing $[-1, :, :]$ selects the last matrix embedding from the ViT's output tensor. The dimensionality of $z_t$ matches that of the input exemplar and query embeddings (i.e., padded token length $\times$ token embedding dimension).

**Next-element Selection.** With this context representation $z_t$, the policy selects the next element by matching $z_t$ against the embeddings of all available candidates. In this context, inspired by

distance-based methods in vision tasks (Mensink et al., 2013; Dong et al., 2015), we use the squared Frobenius norm, $\| \cdot \|_F^2$, as a natural distance metric between these 2D matrix representations. These distances are then converted into a categorical probability distribution using a softmax function. The action space at step $t$ includes all exemplars not yet chosen, $(\mathcal{E} \setminus \tau_{<t})$, plus a special End-of-Sequence (EOS) signal, $e_{[\text{EOS}]}$, which allows the policy to terminate the sequence dynamically.

The complete probability distribution for our policy $\pi_\theta$ selecting the next element $e$ is given by:

$$\pi_\theta(e \mid x, \tau_{<t}) := \frac{\exp\big(-\gamma \cdot \|z_t - e\|_F^2\big)}{\sum_{e' \in (\mathcal{E} \setminus \tau_{<t}) \cup \{e_{[\text{EOS}]}\}} \exp\big(-\gamma \cdot \|z_t - e'\|_F^2\big)}, \quad \forall e \in (\mathcal{E} \setminus \tau_{<t}) \cup \{e_{[\text{EOS}]}\}. \quad (13)$$

The temperature parameter $\gamma > 0$ controls the sharpness of the distribution; a higher $\gamma$ makes the selection more deterministic by favoring candidates with the smallest distance, while a lower $\gamma$ encourages more exploration. The final element $e_{i_t}$ is then sampled from this distribution.

### B.2.2 EOS SIGNAL EMBEDDING

Analogous to the "EOS" token for generation termination in language modeling (Newman et al., 2020), we also formulate an "end-of-sequence" (EOS) embedding $e_{[\text{EOS}]}$ to serve as an ending signal for exemplar selection, when policy model $\pi_\theta$ considers the generated exemplar sequence is good enough to achieve strong performance given the query $x$. Here, inspired by existing works on embedding initialization (Snell et al., 2017; Dobler & de Melo, 2023; Mundra et al., 2024), we set the embedding $e_{[\text{EOS}]}$ as the average exemplar embedding $e_{[\text{EOS}]} \leftarrow \lambda + \frac{1}{|\mathcal{E}|} \sum_{e \in \mathcal{E}} e$ with a small random perturbation $\lambda$. For our experiments, regarding the random perturbation $\lambda$ in EOS signal embedding, we let $\lambda$ be a random matrix, whose elements are individually sampled from zero-mean Gaussian distribution with standard deviation $0.01$. Complementary parameter study for $\lambda$ is also included in Appendix C.2.

### B.2.3 AGGREGATE METRIC REWARD

To obtain a fine-grained training signal, when dealing with possibly low-granularity evaluation metrics $L(\cdot, \cdot)$ (e.g., binary rewards) when plugged into our formulation from Eq. 5, we propose combining feedback by averaging the base metric outcomes over multiple input queries. Analogous techniques are commonly applied in reinforcement learning works, particularly for sparse reward settings (Andrychowicz et al., 2017; Florensa et al., 2018). Specifically, given a small collection of query-answer pairs $\mathcal{D}_{\text{aggr}} = \{(x_i, y_i)\}_{i \in [|\mathcal{D}_{\text{aggr}}|]}$ sampled from the validation data (line 3, Alg. 1), we construct an aggregate query context $\bar{x}$, defined as $\bar{x} = \frac{1}{|\mathcal{D}_{\text{aggr}}|} \sum_{(x,y) \in \mathcal{D}_{\text{aggr}}} x$, inspired by data augmentation techniques (Zhang et al., 2018). Then, the aggregated query $\bar{x}$ is applied to generate the corresponding trajectory collection $\mathcal{T}$ (lines 4-7, Alg. 1). In this context, for each trajectory $\tau \in \mathcal{T}$ generated based on the aggregate query $\bar{x}$, its reward is defined as:

$$r(\bar{x}, \tau) := \frac{1}{|\mathcal{D}_{\text{aggr}}|} \sum_{(x', y') \in \mathcal{D}_{\text{aggr}}} L\big(y', \text{LLM}(x'; \tau)\big).$$

This formulation averages the base metric $L$ over collection $\mathcal{D}_{\text{aggr}}$, yielding a smoother estimate of the trajectory $\tau$'s performance. The resulting aggregated query $\bar{x}$, along with the generated trajectory collection $\mathcal{T}$ and their associated trajectory rewards, will subsequently be used for policy training. In all our experiments, we set $|\mathcal{D}_{\text{aggr}}| = 5$, aggregating feedback from five individual queries, to enhance reward granularity while maintaining computational efficiency.

**Validation data.** For our experiments, the policy $\pi_\theta$ is trained using a reward signal derived from a validation set of query-answer pairs $(x, y)$, which is kept separate from the candidate exemplar pool $\mathcal{E}$. This separation is a crucial methodological control, standard in the field (Zhang et al., 2022; Chen et al., 2024b; Wu et al., 2024), for two reasons. First, it prevents the policy from overfitting to a trivial *lookup* strategy. Second, by using the validation set *only* to generate a scalar reward signal, we compel the model to learn a generalizable selection skill.

### B.3 EXPERIMENT IMPLEMENTATION DETAILS

For our few-shot in-context demonstration selection experiments, each task is associated with $|\mathcal{E}| = 100$ candidate exemplars and an additional 100 validation samples (query-answer pairs),

which are distinct from the exemplar set $\mathcal{E}$. Analogously, we use another separate collection of 400 query-answer pairs as the testing dataset, on which the performance of AUTOSELECT and all baselines is evaluated and reported in our results. The candidate exemplars, validation samples, and testing samples are kept identical for AUTOSELECT and all baseline methods. For AUTOSELECT, we set the regularization coefficient to $\beta = 0.01$. A linear scheduler is applied to the temperature parameter $\gamma$, starting from $\gamma = 0.1$ and increasing linearly to $\gamma = 1$ over the first 200 episodes. The number of rollouts per episode is set to $K = 3$. Our policy model is trained over 400 episodes in a multi-episode training process. In each episode, we perform $K$ trajectory rollouts as indicated in Alg. 1. We set the replay buffer capacity to 50 and sample a small batch size of $|\widehat{\mathcal{B}}| = 10$ for each episode (line 9, Alg. 1), while updating the buffer using a FIFO (First-In, First-Out) strategy to discard outdated information.

To ensure consistency in input length, we pad all exemplars and input queries to a maximum of 320 tokens. For all our experiments, we use the AdamW optimizer (Loshchilov & Hutter, 2019) with the learning rate selected from $\{10^{-5}, 10^{-6}\}$. For our ViT-based policy model, input states are divided into square patches of size 32. The model consists of 4 Transformer blocks, each with an MLP dimension of 512, and 6 attention heads with a head dimension of 64. The output dimensionality of our ViT will consequently match the shape of the query and exemplar embedding matrices, as described in Subsec. 4.1. All experiments are conducted on a Linux server with Intel Xeon CPU and NVIDIA A100 GPUs.

| Task | Descriptions & Query-answer Examples |
|---|---|
| AGNews | A collection of news article titles and descriptions categorized into topics, and it is being used for text classification tasks. |
| | **Input**: No Need for OPEC to Pump More-Iran Gov TEHRAN (Reuters) - OPEC can do nothing to douse scorching oil prices when markets are already oversupplied by 2.8 million barrels per day (bpd) of crude, Iran's OPEC governor said Saturday, warning that prices could fall sharply. |
| | **output**: Business. |
| Amazon | The Amazon dataset contains product reviews from Amazon, including ratings and review text, which are utilized for sentiment analysis and recommender system development. |
| | **Input**: This sound track was beautiful! It paints the senery in your mind so well I would recomend it even to people who hate vid. game music! I have played the game Chrono Cross but out of all of the games I have ever played it has the best music! It backs away from crude keyboarding and takes a fresher step with grate guitars and soulful orchestras. It would impress anyone who cares to listen. |
| | **output**: Positive. |
| SST-2 | SST-2 includes sentence samples extracted from movie reviews, annotated with sentiment labels for sentiment analysis. |
| | **Input**: For those moviegoers who complain that "they don't make movies like they used to anymore." |
| | **output**: Positive. |
| Trec | Trec dataset involves fact-based questions labeled with semantic categories, designed for question classification evaluations. |
| | **Input**: How did serfdom develop in and then leave Russia ? |
| | **output**: Abbreviation. |
| Winowhy | The objective is to evaluate reasoning ability in answering Winograd Schema Challenge questions. |
| | **Input**: The city councilmen refused the demonstrators a permit because they feared violence. The 'they' refers to the city councilmen because The demonstrators advocated violence. |
| | **Output**: Correct. |
| Hyperbaton | The objective is to order adjectives correctly in English sentences. |
| | **Input**: Which sentence has the correct adjective order: a "small Iranian computer" b "Iranian small computer" ? |
| | **Output**: a. |
| Epistemic_reasoning | The objective is to determine whether one sentence entails the next. |
| | **Input**: Premise: James understands that Charles thinks that three children hold a boy's arms down while another boy in a hat shoots a water gun at him. Hypothesis: Charles thinks that James understands that three children hold a boy's arms down while another boy in a hat shoots a water gun at him. |
| | **Output**: Non-entailment. |
| Timedial | The objective is to pick the correct choice for a masked (temporal) span given the dialog context. |
| | **Input**:Which phrase best fits the <MASK> span? Context: A: We need to take the accounts system offline to carry out the upgrade. But don't worry, it won't cause too much inconvenience. We're going to do it over the weekend. B: How long will the system be down for? A: We'll be taking everything offline in about two hours'time. It'll be down for a minimum of twelve hours. If everything goes according to plan, it should be up again by 6 pm on Saturday. B: That's fine. We've allowed <MASK> to be on the safe side. |
| | **Output**: 50 hours. |
| AQuA | The AQuA dataset consists of algebraic and arithmetic word problems in multiple-choice format, requiring both logical reasoning and numerical computation. |
| | **Input**:Two friends plan to walk along a 43-km trail, starting at opposite ends of the trail at the same time. If Friend P's rate is 15% faster than Friend Q's, how many kilometers will Friend P have walked when they pass each other? |
| | **Output**: 23. |

Table 3: Task descriptions and exemplary query-answer pairs.

## C    COMPLEMENTARY EMPIRICAL RESULTS

**Outline.** Due to strict page constraints in the main body, we include additional experiments in this Appendix section. The contents are organized as follows: (1) [Subsec. C.1] Experimental results of AUTOSELECT across diverse task-solving LLMs of various families and scales, demonstrating its broad compatibility. (2) [Subsec. C.2] Impact of hyper-parameters on selection performance, with a parameter study highlighting performance trends, including trajectory length $T$, rollout count $K$, KL coefficient $\beta$, and EOS perturbation scale $\lambda$. (3) [Subsec. C.3] Efficiency-performance analysis comparing runtime and accuracy across methods, plus discussions of sequence-length distributions and inference-time cost. (4) [Subsec. C.4] Backbone comparisons with a GPT-2-based variant of AUTOSELECT, validation our policy model architecture implementation choice (Appendix B.2.1); (5) [Subsection C.5] Qualitative examples of selected exemplar sequences, as well as their potential correlations with input queries.

### C.1    COMPARISON ACROSS TASK-SOLVING LLMS OF VARYING SPECIFICATIONS

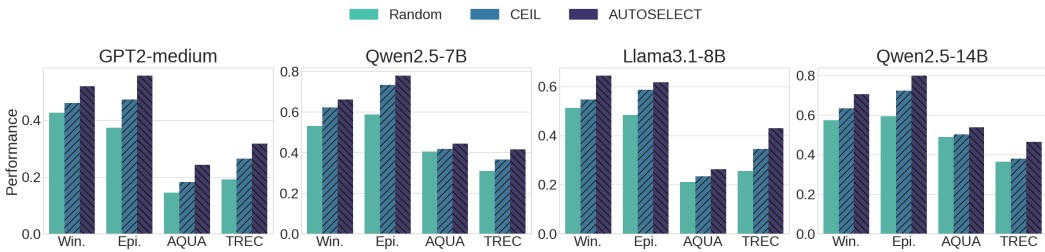

Figure 7: Performance comparison of exemplar selection methods across different task-solving LLMs, across various model families and scales. For abbreviations, "Win." and "Epi." respectively refer to the Winowhy and Epistemic_reasoning datasets.

To evaluate the effectiveness of our AUTOSELECT with different task-solving LLMs with various specifications, we compare performance across four datasets using GPT-2 Medium, LLaMA3.1 8B, and Qwen2.5 models with 7B and 14B parameters. The results, summarized in Fig. 7, provide a comprehensive view of how AUTOSELECT generalizes across varying model scales and architectures.

Across all models and datasets, AUTOSELECT consistently outperforms both random and CEIL baselines, demonstrating its adaptability and effectiveness, regardless of model size or architecture. Here, the improvements are particularly significant on more challenging reasoning tasks such as "Winowhy" and "Epistemic Reasoning", where precise semantic alignment and coherent exemplar context are critical. Compared to CEIL, AUTOSELECT's auto-regressive selection policy can more effectively capture the nuanced dependencies between queries and exemplars, especially when coupled with more capable language models. Another remarkable observation is that GPT-2 Medium and LLaMA3.1-8B show relatively poor performance on the "AQuA" dataset, even with improved exemplar selection. This is likely due to their limited math reasoning capabilities, which constrain the effectiveness of demonstration selection strategies.

### C.2    EFFECT OF HYPER-PARAMETERS ON SELECTION PERFORMANCE

In this subsection, we conduct a parameter study to analyze the effect of key hyper-parameters in AUTOSELECT, including maximum trajectory length $T$, number of trajectory rollouts $K$, and KL regularization coefficient $\beta$. As shown in Fig. 8, results on four datasets are reported as relative performance improvements, over the default settings used in our main experiments (Table 1).

As shown in Fig. 8 (left), increasing the maximum trajectory length $T$ from 4 to 7 consistently improves performance across four tasks, suggesting that longer trajectories can generally offer richer contextual signals for the task-solving LLM. In addition to that, Fig. 8 demonstrates that increasing the number of trajectory rollouts $K$ from 2 to 5 can also consistently improve performance across

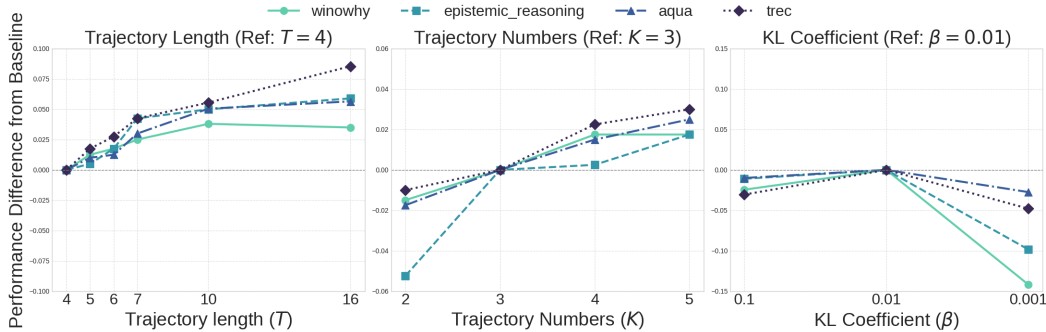

Figure 8: Parameter sensitivity analysis of AUTOSELECT. **Left:** Impact of trajectory maximum length $T$. **Middle:** Effect of the number of trajectory rollouts $K$. **Right:** Influence of KL regularization coefficient $\beta$. All results are reported as relative performance improvements over default settings ($T = 4$, $K = 3$ and $\beta = 0.01$).

four tasks, with the most notable gains observed between $K = 2$ and $K = 4$. This indicates that generating more trajectories can enhance exemplar diversity and improve reward signal quality. On the other hand, Fig. 8 (right) shows the sensitivity to $\beta$, which governs the KL constraint strength during policy updates. We find $\beta = 0.01$ consistently yields the best performance. Here, larger $\beta$ values can result in overly conservative updates, while very small $\beta$ values (e.g., 0.001) can cause policy optimization instability due to insufficient KL regularization.

**Perturbation Scaling $\lambda$.** Recall that we introduce a small random perturbation $\lambda$ to the end-of-sequence (EOS) signal embedding in our instantiation, which is detailed in Appendix B.2. For the random perturbation $\lambda$ in EOS

| Task $\setminus \lambda$ | $\lambda = 0$ | $\lambda = 0.01$ | $\lambda = 0.05$ | $\lambda = 0.1$ |
|---|---|---|---|---|
| Winowhy | $0.641 \pm 0.018$ | $0.657 \pm 0.012$ | $0.647 \pm 0.038$ | $0.639 \pm 0.022$ |
| Aqua | $0.379 \pm 0.004$ | $0.395 \pm 0.002$ | $0.388 \pm 0.008$ | $0.391 \pm 0.011$ |

Table 4: Performance comparison on the Winowhy and AQuA tasks for different $\lambda$ values.

signal embedding, we let $\lambda$ be a random matrix, whose elements are individually sampled from zero-mean Gaussian distribution with standard deviation $\lambda$. Our empirical results suggest this perturbation is beneficial, with the best performance on both Winowhy (0.657) and AQuA (0.395) achieved when setting the noise scaling coefficient to $\lambda = 0.01$. We also investigate the sensitivity of the noise scaling coefficient $\lambda$ for our EOS signal embedding instantiation. From the results below, AutoSelect tends to enjoy stable performance across different choices of $\lambda$.

## C.3   EFFICIENCY-PERFORMANCE ANALYSIS OF SELECTION METHODS

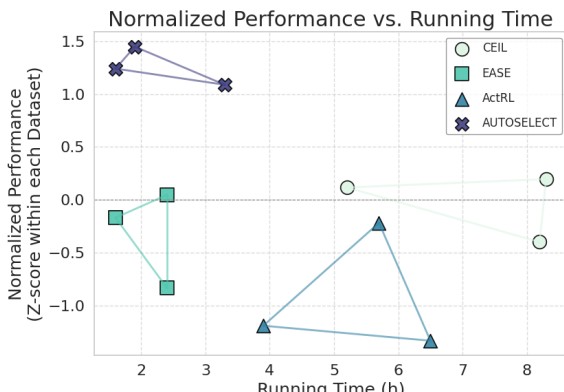

Figure 9: Efficiency comparison of exemplar selection methods in terms of normalized performance versus overall running time (including training, testing, etc.). Each point represents the performance of a method on one of three datasets: "AQuA", "Epistemic_Reasoning", and "Winowhy". Performance is normalized within each dataset using Z-score scaling.

We also conduct an efficiency-performance analysis across three datasets. Fig. 9 visualizes the relationship between running time (in hours) and normalized performance (measured using Z-score within each dataset) for three baseline methods and our proposed framework, AUTOSELECT. As shown in the figure, AUTOSELECT consistently achieves the highest normalized performance across all datasets, while also exhibiting the optimal or near-optimal runtime among all evaluated methods. In contrast, CEIL and ActRL can incur substantially more computational costs, up to 8 hours in some cases. EASE offers relatively efficient runtime, but can lag in normalized performance, particularly on reasoning-heavy tasks like AQuA, due to its use of a static exemplar sequence for all queries within each task rather than query-aware demonstration selection.

**Visualization of Demonstration Selection.** Fig. 10 visualizes the average trajectory lengths per dataset. AUTOSELECT adapts to task-specific characteristics by selecting demonstration sequences of varying lengths, where shorter trajectories are generally selected for less difficult tasks (e.g., "SST-2") and longer ones for challenging reasoning tasks (e.g., "Winowhy"). This highlights AUTOSELECT's ability to tailor its selections to the complexity and reasoning demands of each task, while effectively balancing task performance and LLM inference computational cost by adaptively adjusting the number of exemplars within the context window.

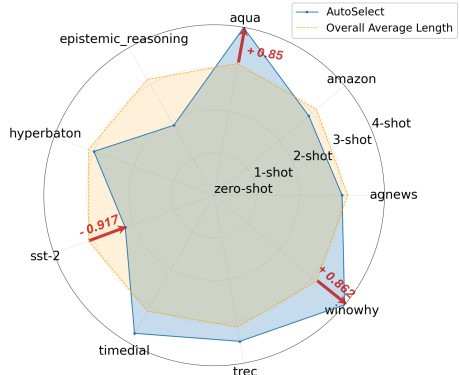

Figure 10: Average trajectory Length on Different Datasets given the maximum length $T = 4$.

**Discussion and Future Work (Long-context RAG).** The computational cost of our auto-regressive approach, which scales with sequence length, is well-aligned with the targeted *few-shot in-context learning* paradigm where performance relies on moderate-length and high-quality demonstration sequences. This efficiency is further enhanced by an integrated *early stopping mechanism* that dynamically learns to terminate the selection process, often yielding minimal, cost-effective sequences as confirmed by the above analysis. While efficiently extending this framework to long-context applications such as Retrieval-Augmented Generation (RAG) (Xu et al., 2023) remains challenging due to the potential computational complexity of auto-regressive exemplar selection, we view this as a promising direction for future work. In particular, we plan to investigate architectural modifications that (i) equip the auto-regressive policy with *sparse or structured attention* mechanisms, such as sparse attention patterns (Child et al., 2019) or block-local and global attention schemes tailored to long documents (Beltagy et al., 2020). This design aims to focus computation on the most relevant portions of the context prefix. Meanwhile, we can (ii) introduce a *context abstraction layer* that compresses the growing prefix into a compact state representation, inspired by hierarchical memory mechanisms for long-range sequence modeling (He et al., 2025). These extensions can help enable AUTOSELECT to practically scale to considerably larger values of $T$ in long-context RAG settings, while AUTOSELECT in its current form is intentionally designed and optimized for the efficiency requirements inherent to our targeted few-shot ICL settings.

**Inference-time Result.** Meanwhile, we can also compare AUTOSELECT (using its learned policy to select maximum $k = 4$ exemplars for each query) against random baseline at a larger scale ($k = 16$). We measure both Test Accuracy and Relative Inference Cost on 400 testing samples of AQuA. From the tabel, AUTOSELECT achieves a higher accuracy with a lower cost. Our method with a maximum of 4 exemplars outperforms a random baseline that uses 16 exemplars. The above computational advantage can intuitively become more significant as the language model scale increases, since larger models are generally more computationally demanding during inference time.

| Method | Exemplars ($k$-shot) | Test Accuracy | Inference Time Cost |
|---|---|---|---|
| Random | 16 | 0.388 | 1 min 53 secs ($\sim$**1.76x**) |
| **AUTOSELECT** | **4** | **0.395** | 1 min 27 secs (**1.0x**) |

Table 5: Inference running time and accuracy comparison on 400 testing samples of AQuA.

Our AUTOSELECT policy training is a justified trade-off. The main "added complexity" of our method lies in its *training phase*, which requires a tiny number of samples from the validation set in

each episode to provide feedback signals (rewards) for learning, which is a *standard and necessary practice* in this line of research (Zhang et al., 2022; Wu et al., 2024; Chen et al., 2024b) to guide policy optimization. Such *one-time and offline cost* of training AUTOSELECT is justified by the considerable downstream savings in inference cost and the superior accuracy it enables. It is an investment that pays off at deployment time by supporting cheaper, faster, and more accurate predictions.

### C.4 EFFECTIVENESS OF VIT BACKBONE CHOICE

To validate our core architectural design choices, using a Vision Transformer (ViT) backbone with 2D matrix representations (Appendix B.2), we conduct an ablation study against a more conventional alternative. For this, we implement a variant of our AUTOSELECT framework where the ViT backbone is replaced with a pre-trained *GPT-2 Medium model*, and the 2D matrix embeddings are replaced with standard *flattened 1D token vectors*.

Crucially, this GPT-2 variant is not a simple zero-shot selector; it is also trained using the exact same auto-regressive paradigm and learning procedure as our proposed framework (as outlined in Alg 1 and Section 4). This ensures that the only differences are the backbone model and the input representation, allowing for a direct and fair comparison of these architectural components.

As shown in Fig. 11, our ViT-based model consistently outperforms the GPT-2 variant, despite the latter having significantly more parameters. This result strongly validates our design, indicating that the synergy between the ViT architecture and 2D matrix representations is more effective at capturing the necessary structural information for this task than a larger, general-purpose transformer operating on flattened data. Notably, both AUTOSELECT variants considerably outperform weaker baselines like ActRL and random selection, underscoring the general effectiveness of our auto-regressive paradigm.

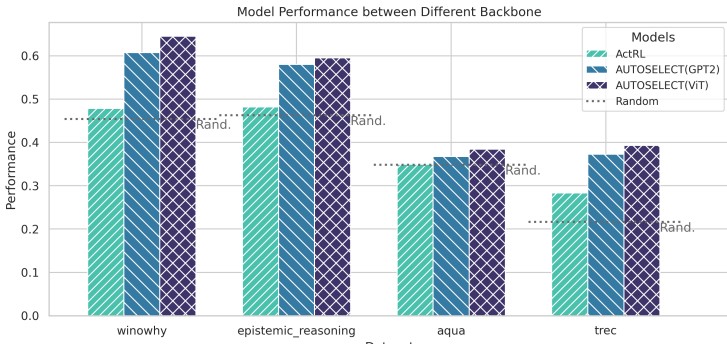

Figure 11: Performance evaluation of GPT-2 Medium as an alternative policy backbone. Horizontal line "Rand." (Random) is reference performance from Table 1.

**Visualization and Discussion.** We also visualize the learned attention over 2D query-exemplar matrices on the AQuA dataset. In Fig. 12, the first figure is one query matrix and the remaining four figures are the selected exemplars. Each heatmap is annotated with its attention weight ("w" value) and its row / column correlations ($r_{\text{row}}$, $r_{\text{col}}$) with the query. We observe that the first three exemplars show strong structural alignment with the query (high $r_{\text{row}}$ / $r_{\text{col}}$ and high weight). This means that the ViT is explicitly matching 2D patterns across both axes, capturing how the "question-reasoning-answer" structure maps onto the exemplars, rather than just comparing token-by-token. By contrast, the fourth exemplar, which is selected last in the auto-regressive trajectory, has a clearly mismatched 2D structure, low (even negative) correlations, and a much smaller attention weight. This indicates that although it is the argmax under the model's scoring at that step, its contribution is effectively down-weighted when aggregating information. This also *suggests that an early-termination (EOS) signal can be beneficial to stop before adding such marginal exemplars.* This kind of spatially coherent and structure-aware matching is hard to realize with a GPT-2-style policy over flattened vectors, where the 2D layout is destroyed and long-range dependencies need to be inferred over a single 1D sequence. Together with the ablation results with the GPT-2 architecture (Fig. 11), these

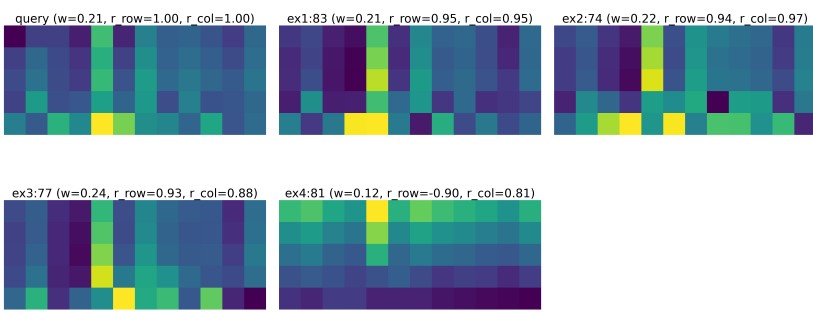

Figure 12: Visualization of the AUTOSELECT policy on 2D attention of AQuA dataset: matrices (first figure: one query; remaining figures: four selected exemplars with attention weights and row / column correlations to the query). AUTOSELECT assigns higher weight ("w" value) to structurally aligned exemplars while down-weighting a structurally mismatched and negatively correlated last exemplar, which highlights the advantage of operating on 2D matrices.

attention maps provide evidence that the ViT policy can effectively leverage the matrix structure to model inter-exemplar relationships.

## C.5 EXAMPLES OF CHOSEN EXEMPLAR SEQUENCE: CORRELATIONS BETWEEN EXEMPLARS CHOSEN AND THE INPUT QUERY.

In this subsection, we present several examples of input queries along with their corresponding exemplar sequences selected by AUTOSELECT, shown in Tables 6-9 below. We also provide insights into the possible rationale behind the exemplar selection outcome.

| | |
|---|---|
| **Input Query** | The ratio of the present ages of a son and his father is 1 : 5 and that of his mother and father is 4 : 5. After 2 years the ratio of the age of the son to that of his mother becomes 3 : 10. What is the present age of the father? |
| **Exemplar 1** | The credit card and a global payment processing companies have been suffering losses for some time now. A well known company recently announced its quarterly results. According to the results, the revenue fell to $48.0 billion from $69.0 billion, a year ago. By what percent did the revenue fall? |
| **Exemplar 2** | Four friends, Peter, John, Quincy, and Andrew, are pooling their money to buy a $1600 item. Peter has twice as much money as John. Quincy has $40 more than Peter. Andrew has 10% more than Quincy. If put all their money together and spend the $1600, they will have $14 left. How much money does Peter have? |
| **Exemplar 3** | Lagaan is levied on the 60 percent of the cultivated land. The revenue department collected total Rs. 3,74,000 through the lagaan from the village of Mutter. Mutter, a very rich farmer , paid only Rs.480 as lagaan. The percentage of total land of Mutter over the total taxable land of the village is: |
| **Exemplar 4** | An exam consists of 8 true/false questions. Brian forgets to study, so he must guess blindly on each question. If any score above 60% is a passing grade, what is the probability that Brian passes? |

Table 6: Demonstration of one exemplary input query and corresponding four chosen exemplars for AQuA dataset. These exemplars are particularly effective because they all involve percentage or ratio information directly correlating with the input query, while simultaneously presenting diverse problem-solving structures. This combination of relevant numerical concepts across various contexts also reinforces the multi-step relational reasoning needed for the input query.

| Input Query | What is the capital of Yugoslavia ? |
| --- | --- |
| Exemplar 1 | What is the address for the main government office in Rome , Italy ? |
| Exemplar 2 | What happened to Pompeii ? |
| Exemplar 3 | How many zip codes are there in the U.S. ? |
| Exemplar 4 | Where did Indian Pudding come from ? |

Table 7: Demonstration of one exemplary input query and corresponding four chosen exemplars for Trec dataset. This exemplar set is particularly beneficial for the query "What is the capital of Yugoslavia?", because the chosen exemplars directly align with the query's need for a specific location-based factual answer, sharing the same broad category of "location". Meanwhile, the remaining exemplars seek different types of information, which helps clarify the query's intent by highlighting its focus on retrieving a geographical entity. This makes the chosen exemplar sequence more instructive than a random or less targeted selection.

| Input Query | The delivery truck zoomed by the school bus because it was going so fast. The 'it' refers to the delivery truck because The delivery bus was faster. |
| --- | --- |
| Exemplar 1 | The table won't fit through the doorway because it is too narrow. The 'it' refers to the doorway because The doorway won't fit through the table because it is too wide. |
| Exemplar 2 | The dog chased the cat, which ran up a tree. It waited at the top. The 'It' refers to the cat because It waited at the top.The dog chased the cat, which ran up a tree. |
| Exemplar 3 | Tom said "Check" to Ralph as he took his bishop. The 'his' refers to ralph because that's the only way we can understand it in the game. |
| Exemplar 4 | Fred was supposed to run the dishwasher, but he put it off, because he wanted to watch TV. But the show turned out to be boring, so he changed his mind and turned it on. The 'it' refers to the dishwasher because it has a TV, but it's possible that the 'it' is the washing machine because of. |

Table 8: Demonstration of one exemplary input query and corresponding four chosen exemplars for Winowhy dataset. This exemplar set is particularly insightful, because it reflects the input query's core reliance on comparative reasoning, such as a truck being "faster" or a table being "too wide" versus a doorway being "too narrow" in Exemplar 1. This helps determine pronoun reference based on contrasting attributes or actions. The diverse comparison contexts across the exemplars, along with Winowhy's characteristic (as seen in Exemplars 1, 3, and 4), help the model resolve the pronoun in the query using similar comparative logic. They also guide the model to produce reasoning that aligns with the dataset's common patterns.

| | |
|---|---|
| **Input Query** | Premise: Charles assumes that Isabella sees that a man is resting in a small stream with a hat over his head while the little waterfall is pouring in the background. Hypothesis: Charles assumes that a man is resting in a small stream with a hat over his head while the little waterfall is pouring in the background. |
| **Exemplar 1** | Premise: John thinks that Isabella knows that boy in green pajamas play with his toy while his mother sits on the couch and watches him. Hypothesis: John thinks that boy in green pajamas play with his toy while his mother sits on the couch and watches him. |
| **Exemplar 2** | Premise: John sees that a basketball player in white in red goes up for the shot as two defensive men in red jump up to block the shot. Hypothesis: John sees that a basketball player is trying to make a shot while two players try to block it. |
| **Exemplar 3** | Premise: Charles believes that Evelyn learns that a person dressed in winter clothes poses with a snowman surrounded by snow covered landscape. Hypothesis: Evelyn learns that a person dressed in winter clothes poses with a snowman surrounded by snow covered landscape. |
| **Exemplar 4** | Premise: Taylor suspects that Olivia thinks that a man on one miniature train passes a circular object to a man on another train as they pass by. Hypothesis: Taylor suspects that a man on one miniature train passes a circular object to a man on another train as they pass by. |

Table 9: Demonstration of one exemplary input query and corresponding four chosen exemplars for Epistemic_reasoning dataset. The above example illustrates how contrasting exemplars with similar structures can support epistemic reasoning in language models. For instance, one exemplar ("John thinks Isabella knows...") mirrors the query's logic, and provides proper reference. Meanwhile, another exemplar ("Taylor suspects Olivia...") applies a similar structural transformation but leads to an incorrect inference. This contrast, paired with an additional exemplar illustrating a different type of invalid simplification, provides valuable reference for the LLM's inference process.

# D  EQUIVALENCE OF POLICY-LEVEL CROSS-ENTROPY (CE) LOSS MINIMIZATION AND INDUCED PLACKETT-LUCE (PL) RANKING OPTIMIZATION

In this section, we provide a detailed description and proof of Theorem 4.2 from the main body, which establishes the connection between minimizing the Cross-Entropy loss and optimizing the Plackett-Luce (PL) ranking, conditioned on a collected trajectory collection $\mathcal{T}$. Subsequently, we can have the following Theorem D.1 that corresponds to Theorem 4.2 in the main body.

---

**Theorem D.1.** *Suppose $\pi^*(\cdot|\boldsymbol{x})$ and $\pi_\theta(\cdot|\boldsymbol{x})$ are two probability distributions induced by two policies $\pi^*, \pi_\theta$ respectively. Let $\mathcal{T}$ be a non-empty finite collection of trajectories such that the total probability masses $Z^* = \sum_{\tau \in \mathcal{T}} \pi^*(\tau|\boldsymbol{x})$ and $Z_\theta = \sum_{\tau \in \mathcal{T}} \pi_\theta(\tau|\boldsymbol{x})$ are strictly positive. Define the conditional distributions over $\mathcal{T}$ as $\pi^*(\tau|\mathcal{T}, \boldsymbol{x}) = \pi^*(\tau|\boldsymbol{x})/Z^*$ and $\pi_\theta(\tau|\mathcal{T}, \boldsymbol{x}) = \pi_\theta(\tau|\boldsymbol{x})/Z_\theta, \forall \tau \in \mathcal{T}$. The probability factor is denoted by $\epsilon := \min\{\pi^*(\tau_{\max}|\mathcal{T}, \boldsymbol{x}), \pi_\theta(\tau_{\max}|\mathcal{T}, \boldsymbol{x})\}$ where $\tau_{\max} = \arg\max_{\tau' \in \mathcal{T}} |\log \frac{\pi^*(\tau'|\mathcal{T}, \boldsymbol{x})}{\pi_\theta(\tau'|\mathcal{T}, \boldsymbol{x})}|$. Then, the absolute difference between PL ranking models can be bounded by*

$$\max_\sigma \left| P_{\pi^*}(\sigma| \mathcal{T}, \boldsymbol{x}) - P_{\pi_\theta}(\sigma| \mathcal{T}, \boldsymbol{x}) \right| \leq \frac{2\beta \cdot |\mathcal{T}|}{\epsilon} \cdot \sqrt{2 \cdot (\mathcal{L}_{CE}^{\mathcal{T}}(\pi^*, \pi_\theta) - \mathcal{L}_{CE}^{\mathcal{T}}(\pi^*, \pi^*))}, \quad (14)$$

*where $\mathcal{L}_{CE}^{\mathcal{T}}(\pi_1, \pi_2)$ refers to the Cross-Entropy (CE) loss given target policy $\pi_1$ and trainable policy $\pi_2$ over trajectory collection $\mathcal{T}$.*

---

*Proof.* To begin with, let $f(p) = \log(p)$ for $p > 0$. By the Mean Value Theorem, if $f$ is continuous on $[a, b]$ and differentiable on $(a, b)$, then there exists $c \in (a, b)$ such that

$$f(b) - f(a) = f'(c)(b - a) \quad (15)$$

Based on the definition from Lemma D.2, by denoting $a = \pi_\theta(\tau|\mathcal{T}, \boldsymbol{x})$ and $b = \pi^*(\tau|\mathcal{T}, \boldsymbol{x})$, we have

$$\log(\pi^*(\tau|\mathcal{T}, \boldsymbol{x})) - \log(\pi_\theta(\tau|\mathcal{T}, \boldsymbol{x})) = \frac{1}{c}(\pi^*(\tau|\mathcal{T}, \boldsymbol{x}) - \pi_\theta(\tau|\mathcal{T}, \boldsymbol{x}))$$

$$\log \frac{\pi^*(\tau|\mathcal{T}, \boldsymbol{x})}{\pi_\theta(\tau|\mathcal{T}, \boldsymbol{x})} = \frac{\pi^*(\tau|\mathcal{T}, \boldsymbol{x}) - \pi_\theta(\tau|\mathcal{T}, \boldsymbol{x})}{c}$$

Since $c$ is between $\pi_\theta(\tau|\mathcal{T}, \boldsymbol{x})$ and $\pi^*(\tau|\mathcal{T}, \boldsymbol{x})$, we know $c \geq \min\{\pi_\theta(\tau|\mathcal{T}, \boldsymbol{x}), \pi^*(\tau|\mathcal{T}, \boldsymbol{x})\}$. As $\frac{1}{c}$ decreases with increasing $c$, we have

$$\left|\log \frac{\pi^*(\tau|\mathcal{T}, \boldsymbol{x})}{\pi_\theta(\tau|\mathcal{T}, \boldsymbol{x})}\right| = \left|\frac{\pi^*(\tau|\mathcal{T}, \boldsymbol{x}) - \pi_\theta(\tau|\mathcal{T}, \boldsymbol{x})}{c}\right|$$

$$\leq \frac{|\pi^*(\tau|\mathcal{T}, \boldsymbol{x}) - \pi_\theta(\tau|\mathcal{T}, \boldsymbol{x})|}{\min\{\pi^*(\tau|\mathcal{T}, \boldsymbol{x}), \pi_\theta(\tau|\mathcal{T}, \boldsymbol{x})\}}.$$

Next, by applying Pinsker's inequality, for any distributions $p$ and $q$, we have $\max_x |p(x) - q(x)| \leq \sqrt{2 \cdot D_{KL}(p||q)}$. In this context, for policies conditioned on trajectories $\mathcal{T}$, denote $\epsilon := \min\{\pi^*(\tau_{\max}|\mathcal{T}, \boldsymbol{x}), \pi_\theta(\tau_{\max}|\mathcal{T}, \boldsymbol{x})\}$ where $\tau_{\max} = \arg\max_{\tau' \in \mathcal{T}} \left|\log \frac{\pi^*(\tau'|\mathcal{T}, \boldsymbol{x})}{\pi_\theta(\tau'|\mathcal{T}, \boldsymbol{x})}\right|$. Then, with the conditional distributions over $\mathcal{T}$ as $P(\tau) := \pi^*(\tau|\mathcal{T}, \boldsymbol{x})$ and $Q(\tau) := \pi_\theta(\tau|\mathcal{T}, \boldsymbol{x})$, we will have

$$\max_{\tau \in \mathcal{T}} \left|\log \frac{\pi^*(\tau|\boldsymbol{x})}{\pi_\theta(\tau|\boldsymbol{x})}\right| \leq \frac{\sqrt{2 D_{\mathrm{KL}}(P \| Q)}}{\epsilon}. \quad (16)$$

Combining the above derivation with the property $D_{\mathrm{KL}}(P \| Q) = \mathcal{L}_{\mathrm{CE}}^{\mathcal{T}}(\pi^*, \pi_\theta) - \mathcal{L}_{\mathrm{CE}}^{\mathcal{T}}(\pi^*, \pi^*)$, together with the conclusion from Lemma D.2, will complete the proof.

$\square$

**Discussion.** The above theorem suggests that minimizing the CE loss in terms of the generated trajectory collection $\mathcal{T}$ can serve as a feasible training objective, for achieving optimal PL ranking results. Furthermore, we can consider that the concentratability condition holds for trajectories and policy models, where analogous conditions are commonly adopted in existing reinforcement learning analyses (e.g., (Chen & Jiang, 2019; Hong et al., 2024)), in order to facilitate theoretical analysis and ensure the stability of policy optimization. Here, if we have $\|\pi^*(\tau|\mathcal{T}, \boldsymbol{x})/\pi_\theta(\tau|\mathcal{T}, \boldsymbol{x})\|_\infty \le \epsilon'$ over trajectories $\tau \in \mathcal{T}$, we can further redefine with its upper bound $\epsilon := \min\{\pi^*(\tau_{\max}|\mathcal{T}, \boldsymbol{x}), \pi^*(\tau_{\max}|\mathcal{T}, \boldsymbol{x})/\epsilon'\}$, which subsequently makes the probability factor $\epsilon$ independent from $\pi_\theta$. This will consequently make the upper bound on the RHS of Eq. 14 decreasing monotonically along with the CE loss, and make the upper bound independent from the policy $\pi_\theta$. Note that this concentratability condition is also realizable in practice, for example, by incorporating exploration techniques such as epsilon-greedy (Dann et al., 2022) into the trajectory generation process.

> **Lemma D.2.** *Let $\pi^*(\cdot|\boldsymbol{x})$ and $\pi_\theta(\cdot|\boldsymbol{x})$ be two policy probability distributions. Let $\mathcal{T}$ be a non-empty finite subset such that the total probability masses $Z^* = \sum_{\tau \in \mathcal{T}} \pi^*(\tau|\boldsymbol{x})$ and $Z_\theta = \sum_{\tau \in \mathcal{T}} \pi_\theta(\tau|\boldsymbol{x})$ are strictly positive. Define the conditional distributions over $\mathcal{T}$ as $\pi^*(\tau|\mathcal{T}, \boldsymbol{x}) = \pi^*(\tau|\boldsymbol{x})/Z^*$ and $\pi_\theta(\tau|\mathcal{T}, \boldsymbol{x}) = \pi_\theta(\tau|\boldsymbol{x})/Z_\theta, \forall \tau \in \mathcal{T}$. For any permutation $\sigma$ of the trajectory collection $\mathcal{T}$, the absolute difference, between the probabilities assigned by the Plackett-Luce (PL) ranking models induced by policies $\pi^*$ and $\pi_\theta$, can be bounded by*
>
> $$|P_{\pi^*}(\sigma|\mathcal{T}, \boldsymbol{x}) - P_{\pi_\theta}(\sigma|\mathcal{T}, \boldsymbol{x})| \le 2\beta \cdot |\mathcal{T}| \cdot \max_{\tau' \in \{\tau_1, \ldots, \tau_{|\mathcal{T}|}\}} \left|\log \frac{\pi^*(\tau'|\mathcal{T}, \boldsymbol{x})}{\pi_\theta(\tau'|\mathcal{T}, \boldsymbol{x})}\right|. \quad (17)$$
>
> *where $|\mathcal{T}|$ is the cardinality of trajectory collection $\mathcal{T}$.*

*Proof.* Based on the definition of the Plackett-Luce (PL) model induced by a policy model $\pi$, due to the shift-invariance property of the softmax function in Lemma E.2, we will have

$$P_\pi(\sigma|\mathcal{T}, \boldsymbol{x}) = \prod_{i=1}^{|\mathcal{T}|} \frac{\exp\left(\beta \log \frac{\pi(\tau_{\sigma(i)}|\boldsymbol{x})}{\pi_{\text{old}}(\tau_{\sigma(i)}|\boldsymbol{x})}\right)}{\sum_{j=i}^{|\mathcal{T}|} \exp\left(\beta \log \frac{\pi(\tau_{\sigma(j)}|\boldsymbol{x})}{\pi_{\text{old}}(\tau_{\sigma(j)}|\boldsymbol{x})}\right)}$$

$$= \prod_{i=1}^{|\mathcal{T}|} \frac{\exp\left(\beta \log \frac{\pi(\tau_{\sigma(i)}|\boldsymbol{x})/\sum_{\tau' \in \mathcal{T}} \pi(\tau'|\boldsymbol{x})}{\pi_{\text{old}}(\tau_{\sigma(i)}|\boldsymbol{x})}\right)}{\sum_{j=i}^{|\mathcal{T}|} \exp\left(\beta \log \frac{\pi(\tau_{\sigma(j)}|\boldsymbol{x})/\sum_{\tau' \in \mathcal{T}} \pi(\tau'|\boldsymbol{x})}{\pi_{\text{old}}(\tau_{\sigma(j)}|\boldsymbol{x})}\right)} = \prod_{i=1}^{|\mathcal{T}|} \frac{\exp\left(\beta \log \frac{\pi(\tau_{\sigma(i)}|\mathcal{T}, \boldsymbol{x})}{\pi_{\text{old}}(\tau_{\sigma(i)}|\boldsymbol{x})}\right)}{\sum_{j=i}^{|\mathcal{T}|} \exp\left(\beta \log \frac{\pi(\tau_{\sigma(j)}|\mathcal{T}, \boldsymbol{x})}{\pi_{\text{old}}(\tau_{\sigma(j)}|\boldsymbol{x})}\right)}. \quad (18)$$

Let us denote $s_i(\pi) = \exp\left(\beta \log \frac{\pi(\tau_{\sigma(i)}|\mathcal{T}, \boldsymbol{x})}{\pi_{\text{old}}(\tau_{\sigma(i)}|\mathcal{T}, \boldsymbol{x})}\right) = \left(\frac{\pi(\tau_{\sigma(i)}|\mathcal{T}, \boldsymbol{x})}{\pi_{\text{old}}(\tau_{\sigma(i)}|\mathcal{T}, \boldsymbol{x})}\right)^\beta$, and also denote $Z_i(\pi) = \sum_{j=i}^{|\mathcal{T}|} s_j(\pi)$. Then, the PL model probability can be written as

$$P_\pi(\sigma|\mathcal{T}, \boldsymbol{x}) = \prod_{i=1}^{|\mathcal{T}|} \frac{s_i(\pi)}{Z_i(\pi)} \quad (19)$$

**Decomposition of the objective.** We then focus on how the probability differs between the optimal policy $\pi^*$ and our learnable policy $\pi_\theta$. Given four positive values $a, b, c, d > 0$, we begin by writing

$$\frac{a}{b} - \frac{c}{d} = \frac{ad - bc}{bd}.$$

One way to split the numerator is to write $ad - bc = d(a - c) + c(d - b)$. Taking absolute values and applying the triangle inequality yields

$$\left|\frac{a}{b} - \frac{c}{d}\right| \le \frac{d|a - c|}{bd} + \frac{c|d - b|}{bd} = \frac{|a - c|}{b} + \frac{c}{bd}|b - d|.$$

Alternatively, we can write $ad - bc = a(d - b) + b(a - c)$, and we can similarly obtain

$$\left|\frac{a}{b} - \frac{c}{d}\right| \le \frac{|a - c|}{d} + \frac{a}{bd}|b - d|.$$

Taking the minimum of these two bounds gives the final inequality:

$$\left|\frac{a}{b} - \frac{c}{d}\right| \le \min\left\{\frac{|a-c|}{b} + \frac{c}{bd}|b-d|, \quad \frac{|a-c|}{d} + \frac{a}{bd}|b-d|\right\}.$$

The above derivation shows that by splitting the numerator in two different ways and applying the triangle inequality, we can obtain two valid bounds, and taking their minimum provides the upper bound.

Subsequently, we can use the above inequality to bound the difference in each factor of the product. In particular, for each $i \in |\mathcal{T}|$, we can formulate the bound as

$$\left|\frac{s_i(\pi^*)}{Z_i(\pi^*)} - \frac{s_i(\pi_\theta)}{Z_i(\pi_\theta)}\right| \le \begin{cases} \dfrac{|s_i(\pi^*) - s_i(\pi_\theta)|}{Z_i(\pi^*)} + \dfrac{s_i(\pi_\theta)}{Z_i(\pi^*)\,Z_i(\pi_\theta)}\,|Z_i(\pi^*) - Z_i(\pi_\theta)|, & \text{if } s_i(\pi^*) \ge s_i(\pi_\theta), \\[3mm] \dfrac{|s_i(\pi^*) - s_i(\pi_\theta)|}{Z_i(\pi_\theta)} + \dfrac{s_i(\pi^*)}{Z_i(\pi^*)\,Z_i(\pi_\theta)}\,|Z_i(\pi^*) - Z_i(\pi_\theta)|, & \text{if } s_i(\pi^*) < s_i(\pi_\theta). \end{cases}$$

$$(20)$$

Next, without loss of generality, we consider $s_i(\pi^*) \ge s_i(\pi_\theta)$ for the proof below while applying the first inequity in Eq. 20. We also note the other case $s_i(\pi^*) < s_i(\pi_\theta)$ can also be readily proved by following an analogous procedure, and by alternatively applying the second inequity in Eq. 20. We then proceed to bound $|s_i(\pi^*) - s_i(\pi_\theta)|$ and $|Z_i(\pi^*) - Z_i(\pi_\theta)|$.

To begin with, **(1) for the first term**, we set $a = \frac{\pi^*(\tau_{\sigma(i)}|\mathcal{T},\boldsymbol{x})}{\pi_{\text{old}}(\tau_{\sigma(i)}|\mathcal{T},\boldsymbol{x})}$, $b = \frac{\pi_\theta(\tau_{\sigma(i)}|\mathcal{T},\boldsymbol{x})}{\pi_{\text{old}}(\tau_{\sigma(i)}|\mathcal{T},\boldsymbol{x})}$, so that $s_i(\pi^*) = a^\beta$, and $s_i(\pi_\theta) = b^\beta$. By the Mean Value Theorem on $f(x) = x^\beta$, there exists some $\xi$ strictly between $a$ and $b$ such that $a^\beta - b^\beta = \beta\xi^{\beta-1}(a-b)$. Since $\beta > 0$ and $\xi > 0$ (as it's between $a, b > 0$), $\xi^{\beta-1} > 0$. Thus, taking absolute values gives:

$$|a^\beta - b^\beta| = \beta\xi^{\beta-1}|a - b|. \quad (*)$$

Next, applying the Mean Value Theorem to $g(x) = \log x$, there exists some $\eta$ strictly between $a$ and $b$ such that $\log a - \log b = \frac{1}{\eta}(a - b)$. Since $\eta > 0$:

$$|a - b| = \eta|\log a - \log b|. \quad (**)$$

Substituting equation $(**)$ into equation $(*)$ yields:

$$|a^\beta - b^\beta| = \beta(\xi^{\beta-1}\eta)|\log a - \log b|.$$

To evaluate the term $\xi^{\beta-1}\eta$, we then apply Cauchy's Mean Value Theorem, such that for $F(x) = x^\beta$ and $G(x) = \log x$, there exists $c$ strictly between $a$ and $b$ such that

$$\frac{F(a) - F(b)}{G(a) - G(b)} = \frac{F'(c)}{G'(c)} \implies \frac{a^\beta - b^\beta}{\log a - \log b} = \frac{\beta c^{\beta-1}}{1/c} = \beta c^\beta.$$

Thus, $\frac{|a^\beta - b^\beta|}{|\log a - \log b|} = \beta c^\beta$ (since $\beta > 0, c^\beta > 0$). Comparing this with our combined expression, we can have that $\xi^{\beta-1}\eta = c^\beta$. Since $c$ is strictly between $a$ and $b$, and the function $h(x) = x^\beta$ is strictly increasing for $x > 0$ (because $\beta > 0$), it follows that $c^\beta < \max\{a^\beta, b^\beta\}$. Therefore, $c^\beta \le \max\{a^\beta, b^\beta\}$. Substituting $\xi^{\beta-1}\eta = c^\beta$ and applying above results, we consequently have

$$|a^\beta - b^\beta| = \beta c^\beta|\log a - \log b| \le \beta\max\{a^\beta, b^\beta\}|\log a - \log b|.$$

Returning to our original notation, we can therefore have

$$|s_i(\pi^*) - s_i(\pi_\theta)| \le \beta\max\{s_i(\pi^*),\, s_i(\pi_\theta)\}\left|\log\frac{\pi^*(\tau_{\sigma(i)} \mid \mathcal{T}, \boldsymbol{x})}{\pi_\theta(\tau_{\sigma(i)} \mid \mathcal{T}, \boldsymbol{x})}\right|.$$

**(2) Similarly, for the second term**, we can have

$$|Z_i(\pi^*) - Z_i(\pi_\theta)| = \left|\sum_{j=i}^{|\mathcal{T}|}(s_j(\pi^*) - s_j(\pi_\theta))\right|$$

$$\le \sum_{j=i}^{|\mathcal{T}|}|s_j(\pi^*) - s_j(\pi_\theta)| \le \beta \cdot \sum_{j=i}^{|\mathcal{T}|}\max\{s_j(\pi^*), s_j(\pi_\theta)\} \cdot \left|\log\frac{\pi^*(\tau_{\sigma(j)}|\mathcal{T}, \boldsymbol{x})}{\pi_\theta(\tau_{\sigma(j)}|\mathcal{T}, \boldsymbol{x})}\right|$$

Let $\delta = \max_{\tau' \in \{\tau_1, \ldots, \tau_{|\mathcal{T}|}\}} \left| \log \frac{\pi^*(\tau'|\mathcal{T}, \boldsymbol{x})}{\pi_\theta(\tau'|\mathcal{T}, \boldsymbol{x})} \right|$, which is related to the maximum log-ratio between the optimal and trainable policies across collected trajectories. Then

$$|s_i(\pi^*) - s_i(\pi_\theta)| \leq \beta \cdot \max\{s_i(\pi^*), s_i(\pi_\theta)\} \cdot \delta$$

$$|Z_i(\pi^*) - Z_i(\pi_\theta)| \leq \beta \cdot \sum_{j=i}^{|\mathcal{T}|} \max\{s_j(\pi^*), s_j(\pi_\theta)\} \cdot \delta$$

Substituting these bounds back

$$\left| \frac{s_i(\pi^*)}{Z_i(\pi^*)} - \frac{s_i(\pi_\theta)}{Z_i(\pi_\theta)} \right| \leq \frac{\beta \cdot \max\{s_i(\pi^*), s_i(\pi_\theta)\} \cdot \delta}{Z_i(\pi^*)}$$

$$+ \frac{s_i(\pi_\theta)}{Z_i(\pi^*)Z_i(\pi_\theta)} \cdot \beta \cdot \sum_{j=i}^{|\mathcal{T}|} \max\{s_j(\pi^*), s_j(\pi_\theta)\} \cdot \delta$$

We can further simplify this bound to

$$\left| \frac{s_i(\pi^*)}{Z_i(\pi^*)} - \frac{s_i(\pi_\theta)}{Z_i(\pi_\theta)} \right| \leq \beta \cdot \delta \cdot \left( \frac{\max\{s_i(\pi^*), s_i(\pi_\theta)\}}{Z_i(\pi^*)} + \frac{s_i(\pi_\theta) \cdot \sum_{j=i}^{|\mathcal{T}|} \max\{s_j(\pi^*), s_j(\pi_\theta)\}}{Z_i(\pi^*)Z_i(\pi_\theta)} \right)$$

Here, since we consider $s_i(\pi^*) \geq s_i(\pi_\theta)$ without loss of generality for the proof, we have

$$\frac{\max\{s_i(\pi^*), s_i(\pi_\theta)\}}{Z_i(\pi^*)} = \frac{s_i(\pi^*)}{Z_i(\pi^*)} \leq 1.$$

Meanwhile, for the second term, we have

$$\frac{s_i(\pi_\theta) \cdot \sum_{j=i}^{|\mathcal{T}|} \max\{s_j(\pi^*), s_j(\pi_\theta)\}}{Z_i(\pi^*)Z_i(\pi_\theta)} = \frac{s_i(\pi_\theta)s_i(\pi^*) + s_i(\pi_\theta) \cdot \sum_{j=i+1}^{|\mathcal{T}|} \max\{s_j(\pi^*), s_j(\pi_\theta)\}}{Z_i(\pi^*)Z_i(\pi_\theta)} \leq 1,$$

where in this expression, each product term in the numerator will appear in the decomposition of the denominator (by appropriately relaxing $s_i(\pi_\theta)$ to $s_i(\pi^*)$ when needed). Thus, we will also have $\frac{s_i(\pi_\theta) \cdot \sum_{j=i}^{|\mathcal{T}|} \max\{s_j(\pi^*), s_j(\pi_\theta)\}}{Z_i(\pi^*)Z_i(\pi_\theta)} \leq 1$.

As we have mentioned previously, note that above results can also be analogously derived when $s_i(\pi^*) < s_i(\pi_\theta)$, by alternatively adopting the second inequity in Eq. 20. As a result, we will have

$$\left| \frac{s_i(\pi^*)}{Z_i(\pi^*)} - \frac{s_i(\pi_\theta)}{Z_i(\pi_\theta)} \right| \leq 2\beta \cdot \delta.$$

Afterwards, recall the full product is given by

$$|P_{\pi^*}(\sigma | \mathcal{T}, \boldsymbol{x}) - P_{\pi_\theta}(\sigma | \mathcal{T}, \boldsymbol{x})| = \left| \prod_{i=1}^{|\mathcal{T}|} \frac{s_i(\pi^*)}{Z_i(\pi^*)} - \prod_{i=1}^{|\mathcal{T}|} \frac{s_i(\pi_\theta)}{Z_i(\pi_\theta)} \right|.$$

We first observe that

$$\prod_{i=1}^{n} A_i - \prod_{i=1}^{n} B_i = \sum_{i=1}^{n} \left( \prod_{j=1}^{i-1} A_j \right) (A_i - B_i) \left( \prod_{j=i+1}^{n} B_j \right).$$

Taking absolute values and applying the triangle inequality will give us $\left| \prod_{i=1}^{n} A_i - \prod_{i=1}^{n} B_i \right| \leq \sum_{i=1}^{n} |A_i - B_i| \left| \prod_{j=1}^{i-1} A_j \right| \left| \prod_{j=i+1}^{n} B_j \right|$. For each index $j$, we will have $A_j, B_j \leq \max\{A_j, B_j\}$, and it follows the fact that $\left| \prod_{j=1}^{i-1} A_j \right| \left| \prod_{j=i+1}^{n} B_j \right| \leq \prod_{j=1, j\neq i}^{n} \max\{A_j, B_j\}$. In this context, we can obtain the following inequality

$$\left| \prod_{i=1}^{n} A_i - \prod_{i=1}^{n} B_i \right| \leq \sum_{i=1}^{n} |A_i - B_i| \prod_{j=1, j\neq i}^{n} \max\{A_j, B_j\}.$$

In our settings, denoting $A_i = \frac{s_i(\pi^*)}{Z_i(\pi^*)}$ and $B_i = \frac{s_i(\pi_\theta)}{Z_i(\pi_\theta)}$, the above derived telescoping-product inequality will consequently lead to

$$|P_{\pi^*}(\sigma|\,\mathcal{T}, \boldsymbol{x}) - P_{\pi_\theta}(\sigma|\,\mathcal{T}, \boldsymbol{x})| \le \sum_{i=1}^{|\mathcal{T}|} \left| \frac{s_i(\pi^*)}{Z_i(\pi^*)} - \frac{s_i(\pi_\theta)}{Z_i(\pi_\theta)} \right| \cdot \prod_{j \ne i} \max\left\{ \frac{s_j(\pi^*)}{Z_j(\pi^*)}, \frac{s_j(\pi_\theta)}{Z_j(\pi_\theta)} \right\}$$

$$\le \sum_{i=1}^{|\mathcal{T}|} 2\beta \cdot \delta \cdot \prod_{j \ne i} 1$$

$$= 2\beta \cdot |\mathcal{T}| \cdot \delta$$

where the first inequality is because each fraction $\frac{s_j(\pi)}{Z_j(\pi)} \le 1$, and their product is also at most 1. Summarizing all the results will give us

$$|P_{\pi^*}(\sigma|\,\mathcal{T}, \boldsymbol{x}) - P_{\pi_\theta}(\sigma|\,\mathcal{T}, \boldsymbol{x})| \le 2\beta \cdot |\mathcal{T}| \cdot \max_{\tau' \in \{\tau_1, \dots, \tau_{|\mathcal{T}|}\}} \left| \log \frac{\pi^*(\tau'|\mathcal{T}, \boldsymbol{x})}{\pi_\theta(\tau'|\mathcal{T}, \boldsymbol{x})} \right|,$$

which completes the proof.

$\square$

# E    EQUIVALENCE OF POLICY MODELS AND INDUCED PLACKETT-LUCE (PL) RANKING MODELS

We consider two policy models: $\pi_\theta$ is a trainable policy model with parameters $\theta$, and $\pi^*$ is the optimal policy model. Recall that for a collection of trajectories $\mathcal{T} := \{\tau_1, \tau_2, \dots, \tau_{|\mathcal{T}|}\}$ and query $\boldsymbol{x}$, the Plackett-Luce (PL) ranking model induced by a policy $\pi$, relative to a previous policy $\pi_{\text{old}}$, is defined as:

$$P_\pi(\sigma|\mathcal{T}, \boldsymbol{x}) = \prod_{i=1}^{|\mathcal{T}|} \frac{\exp\left(\beta \log \frac{\pi(\tau_{\sigma(i)}|\boldsymbol{x})}{\pi_{\text{old}}(\tau_{\sigma(i)}|\boldsymbol{x})}\right)}{\sum_{j=i}^{|\mathcal{T}|} \exp\left(\beta \log \frac{\pi(\tau_{\sigma(j)}|\boldsymbol{x})}{\pi_{\text{old}}(\tau_{\sigma(j)}|\boldsymbol{x})}\right)}, \tag{21}$$

where we have $\sigma$ being a permutation (or ranking) of the indices $\{1, 2, \dots, |\mathcal{T}|\}$, and denote $\tau_{\sigma(i)}$ being the trajectory ranked at position $i$ in permutation $\sigma$. $\frac{\pi(\tau|\boldsymbol{x})}{\pi_{\text{old}}(\tau|\boldsymbol{x})}$ represents the relative preference of policy $\pi$ over the previous policy before updating.

Subsequently, let us denote the two induced PL ranking models by

$$P_{\pi_\theta}(\sigma|\mathcal{T}, \boldsymbol{x}) = \prod_{i=1}^{|\mathcal{T}|} \frac{\exp\left(\beta \log \frac{\pi_\theta(\tau_{\sigma(i)}|\boldsymbol{x})}{\pi_{\text{old}}(\tau_{\sigma(i)}|\boldsymbol{x})}\right)}{\sum_{j=i}^{|\mathcal{T}|} \exp\left(\beta \log \frac{\pi_\theta(\tau_{\sigma(j)}|\boldsymbol{x})}{\pi_{\text{old}}(\tau_{\sigma(j)}|\boldsymbol{x})}\right)}, \quad P_{\pi^*}(\sigma|\mathcal{T}, \boldsymbol{x}) = \prod_{i=1}^{|\mathcal{T}|} \frac{\exp\left(\beta \log \frac{\pi^*(\tau_{\sigma(i)}|\boldsymbol{x})}{\pi_{\text{old}}(\tau_{\sigma(i)}|\boldsymbol{x})}\right)}{\sum_{j=i}^{|\mathcal{T}|} \exp\left(\beta \log \frac{\pi^*(\tau_{\sigma(j)}|\boldsymbol{x})}{\pi_{\text{old}}(\tau_{\sigma(j)}|\boldsymbol{x})}\right)}$$

We will then have the following result on the equivalence between the optimal policy and the optimal PL ranking model. Results from the following Proposition E.1 supports the conclusion from Proposition 4.1.

> **Proposition E.1.** *For a learnable policy $\pi_\theta$, the condition that $P_{\pi^*}(\sigma \mid \mathcal{T}, \boldsymbol{x}) = P_{\pi_\theta}(\sigma \mid \mathcal{T}, \boldsymbol{x})$ holds for any possible trajectory collection $\mathcal{T}$ is equivalent to the policies being identical, i.e., $\pi^* = \pi_\theta$, meaning that $\pi^*(\tau \mid \boldsymbol{x}) = \pi_\theta(\tau \mid \boldsymbol{x})$ for all trajectories $\tau$ given $\boldsymbol{x}$.*

*Proof.* We need to prove both directions of the equivalence, and we will start with the forward direction that optimal policy indicates the optimal PL ranking.

**Forward Direction.** Suppose $\pi_\theta(\tau|\boldsymbol{x}) = \pi^*(\tau|\boldsymbol{x})$ for all $\tau$ and $\boldsymbol{x}$. Then, since the policies are identical, their ratios with respect to the old policy are also identical, leading to

$$\frac{\pi_\theta(\tau|\boldsymbol{x})}{\pi_{\text{old}}(\tau|\boldsymbol{x})} = \frac{\pi^*(\tau|\boldsymbol{x})}{\pi_{\text{old}}(\tau|\boldsymbol{x})} \tag{22}$$

Therefore, with $\beta > 0$, for any permutation $\sigma$:

$$P_{\pi_\theta}(\sigma|\mathcal{T},\boldsymbol{x}) = \prod_{i=1}^{|\mathcal{T}|} \frac{\exp\left(\beta \log \frac{\pi_\theta(\tau_{\sigma(i)}|\boldsymbol{x})}{\pi_{\text{old}}(\tau_{\sigma(i)}|\boldsymbol{x})}\right)}{\sum_{j=i}^{|\mathcal{T}|} \exp\left(\beta \log \frac{\pi_\theta(\tau_{\sigma(j)}|\boldsymbol{x})}{\pi_{\text{old}}(\tau_{\sigma(j)}|\boldsymbol{x})}\right)} = \prod_{i=1}^{|\mathcal{T}|} \frac{\exp\left(\beta \log \frac{\pi^*(\tau_{\sigma(i)}|\boldsymbol{x})}{\pi_{\text{old}}(\tau_{\sigma(i)}|\boldsymbol{x})}\right)}{\sum_{j=i}^{|\mathcal{T}|} \exp\left(\beta \log \frac{\pi^*(\tau_{\sigma(j)}|\boldsymbol{x})}{\pi_{\text{old}}(\tau_{\sigma(j)}|\boldsymbol{x})}\right)}$$

$$= P_{\pi^*}(\sigma|\mathcal{T},\boldsymbol{x})$$

Thus, if the policies are identical, then their induced PL ranking models are also identical.

**Reverse Direction.** Suppose $P_{\pi_\theta}(\sigma \mid \mathcal{T}, \boldsymbol{x}) = P_{\pi^*}(\sigma \mid \mathcal{T}, \boldsymbol{x})$ for all permutations $\sigma$, all finite trajectory sets $\mathcal{T}$, and all queries $\boldsymbol{x}$ (with $\beta > 0$ and $\pi_{\text{old}}(\cdot \mid \boldsymbol{x}) > 0$ on the support). Define the scores

$$s_\pi(\tau \mid \boldsymbol{x}) := \beta \log \frac{\pi(\tau \mid \boldsymbol{x})}{\pi_{\text{old}}(\tau \mid \boldsymbol{x})}.$$

Consider any two trajectories $\tau, \tau'$ and the two-trajectory collection $\mathcal{T} = \{\tau, \tau'\}$. The PL probabilities for the two possible permutations satisfy

$$P_{\pi_\theta}\big((\tau,\tau') \mid \{\tau,\tau'\}, \boldsymbol{x}\big) = P_{\pi^*}\big((\tau,\tau') \mid \{\tau,\tau'\}, \boldsymbol{x}\big), \quad P_{\pi_\theta}\big((\tau',\tau) \mid \{\tau,\tau'\}, \boldsymbol{x}\big) = P_{\pi^*}\big((\tau',\tau) \mid \{\tau,\tau'\}, \boldsymbol{x}\big).$$

By the PL definition on a two-item set $\{\tau, \tau'\}$, the probability of placing $\tau$ first under $\pi$ is

$$P_\pi\big((\tau,\tau') \mid \{\tau,\tau'\}, \boldsymbol{x}\big) = \frac{e^{s_\pi(\tau|\boldsymbol{x})}}{e^{s_\pi(\tau|\boldsymbol{x})} + e^{s_\pi(\tau'|\boldsymbol{x})}} = \frac{1}{1 + \exp\big(s_\pi(\tau' \mid \boldsymbol{x}) - s_\pi(\tau \mid \boldsymbol{x})\big)}.$$

Hence equality of the two top-1 probabilities for $\pi_\theta$ and $\pi^*$ implies equality

$$\frac{P_{\pi_\theta}((\tau,\tau') \mid \{\tau,\tau'\}, \boldsymbol{x})}{P_{\pi_\theta}((\tau',\tau) \mid \{\tau,\tau'\}, \boldsymbol{x})} = \frac{P_{\pi^*}((\tau,\tau') \mid \{\tau,\tau'\}, \boldsymbol{x})}{P_{\pi^*}((\tau',\tau) \mid \{\tau,\tau'\}, \boldsymbol{x})} \quad \Longleftrightarrow \quad e^{s_{\pi_\theta}(\tau|\boldsymbol{x}) - s_{\pi_\theta}(\tau'|\boldsymbol{x})} = e^{s_{\pi^*}(\tau|\boldsymbol{x}) - s_{\pi^*}(\tau'|\boldsymbol{x})}.$$

Taking logarithms yields

$$s_{\pi_\theta}(\tau \mid \boldsymbol{x}) - s_{\pi_\theta}(\tau' \mid \boldsymbol{x}) = s_{\pi^*}(\tau \mid \boldsymbol{x}) - s_{\pi^*}(\tau' \mid \boldsymbol{x}).$$

Fix an arbitrary reference $\tau_0$ and set $\tau' = \tau_0$. The above identity then gives, for every $\tau$,

$$s_{\pi_\theta}(\tau \mid \boldsymbol{x}) - s_{\pi_\theta}(\tau_0 \mid \boldsymbol{x}) = s_{\pi^*}(\tau \mid \boldsymbol{x}) - s_{\pi^*}(\tau_0 \mid \boldsymbol{x}),$$

which is equivalent to the existence of a constant $C(\boldsymbol{x}) := s_{\pi_\theta}(\tau_0 \mid \boldsymbol{x}) - s_{\pi^*}(\tau_0 \mid \boldsymbol{x})$ (independent of $\tau$) such that

$$s_{\pi_\theta}(\tau \mid \boldsymbol{x}) = s_{\pi^*}(\tau \mid \boldsymbol{x}) + C(\boldsymbol{x}), \quad \forall \tau.$$

Equivalently,

$$\beta \log \frac{\pi_\theta(\tau \mid \boldsymbol{x})}{\pi_{\text{old}}(\tau \mid \boldsymbol{x})} = \beta \log \frac{\pi^*(\tau \mid \boldsymbol{x})}{\pi_{\text{old}}(\tau \mid \boldsymbol{x})} + C(\boldsymbol{x}),$$

so dividing by $\beta$ and exponentiating gives

$$\frac{\pi_\theta(\tau \mid \boldsymbol{x})}{\pi_{\text{old}}(\tau \mid \boldsymbol{x})} = \frac{\pi^*(\tau \mid \boldsymbol{x})}{\pi_{\text{old}}(\tau \mid \boldsymbol{x})} e^{C(\boldsymbol{x})/\beta} \quad \Longrightarrow \quad \pi_\theta(\tau \mid \boldsymbol{x}) = e^{C(\boldsymbol{x})/\beta} \pi^*(\tau \mid \boldsymbol{x}).$$

Since both $\pi_\theta$ and $\pi^*$ are probability distributions and will sum to 1 over all possible trajectories, $e^{C(\boldsymbol{x})/\beta} = 1$, which implies $\pi_\theta(\tau|\boldsymbol{x}) = \pi^*(\tau|\boldsymbol{x})$. Therefore, if the induced PL ranking models are identical, then the underlying policy models will also be identical.

$\square$

**Lemma E.2** (Softmax Shift-invariance Property). *Let $\boldsymbol{z}, \boldsymbol{w} \in \mathbb{R}^d$, $d \geq 2$ be two vectors of the same dimension. The softmax outputs are identical, softmax$(\boldsymbol{z})$ = softmax$(\boldsymbol{w})$, if and only if there exists a scalar constant $C \in \mathbb{R}$ such that the inputs differ by a constant shift, i.e., $\boldsymbol{w} = \boldsymbol{z} + C \cdot \mathbf{1}$, where $\mathbf{1}$ is the vector of ones, and the softmax function is defined component-wise as softmax$(\boldsymbol{x})_i = \frac{\exp(x_i)}{\sum_{j=1}^K \exp(x_j)}$ for a position $i$.*

*Proof. (Forward)* Suppose vector $\boldsymbol{w} = \boldsymbol{z} + C \cdot \boldsymbol{1}$ for some constant $C$. Then for any component $i$:

$$\text{softmax}(\boldsymbol{w})_i = \frac{\exp(w_i)}{\sum_{j=1}^{K} \exp(w_j)} = \frac{\exp(z_i + C)}{\sum_{j=1}^{K} \exp(z_j + C)}$$

$$= \frac{\exp(z_i)e^C}{e^C \sum_{j=1}^{K} \exp(z_j)} = \frac{\exp(z_i)}{\sum_{j=1}^{K} \exp(z_j)} = \text{softmax}(\boldsymbol{z})_i.$$

Since this holds for all $i$, $\text{softmax}(\boldsymbol{w}) = \text{softmax}(\boldsymbol{z})$.

*(Backward)* Suppose $\text{softmax}(\boldsymbol{z}) = \text{softmax}(\boldsymbol{w})$. Let $S_z = \sum_j \exp(z_j)$ and $S_w = \sum_j \exp(w_j)$. This condition implies $\frac{\exp(z_i)}{S_z} = \frac{\exp(w_i)}{S_w}$ for all $i$. Since $S_z, S_w > 0$, we rearrange to get

$$\exp(w_i) = \exp(z_i) \cdot (S_w/S_z).$$

Let the positive constant $K = S_w/S_z$. Taking the natural logarithm yields $w_i = \ln(\exp(z_i)K) = z_i + \ln(K)$. Setting the constant $C = \ln(K)$, we have $w_i = z_i + C$ for all $i$. Thus, we will have $\boldsymbol{w} = \boldsymbol{z} + C \cdot \boldsymbol{1}$, which completes the proof.

$\square$

# F  EQUIVALENT REWARD FUNCTIONS AND POLICY INVARIANCE

The theoretical foundation of our optimization procedure relies on a key property of our KL-regularized reinforcement learning (RL) problem: the optimal policy is invariant to certain transformations of the reward function, stated in the following lemma.

> **Lemma F.1** (Equivalent Reward Functions (Rafailov et al., 2024)). *Reward functions $r(\boldsymbol{x}, \tau)$ and $r'(\boldsymbol{x}, \tau)$ are equivalent if and only if $r(\boldsymbol{x}, \tau) - r'(\boldsymbol{x}, \tau) = \zeta(\boldsymbol{x})$, where $\zeta(\cdot)$ is an arbitrary function depending on query $\boldsymbol{x}$. For the RL problem in Eq. 6, these equivalent reward functions induce the same optimal policy $\pi^*$ and a unique optimal PL ranking model $P_{\pi^*}(\sigma \mid \mathcal{T}, \boldsymbol{x})$.*

Lemma F.1 establishes that our learning objective is invariant to any baseline reward adjustment, which is constant across all possible trajectories $\tau$ for a given query $\boldsymbol{x}$. This property is crucial as it ensures that our method learns the true *relative preferences* among demonstration sequences, which is the core of the selection task.

**Intuition.** The invariance property stems directly from the exponential form of the optimal policy solution in Eq. 7. Given $\pi^*(\tau|\boldsymbol{x}) \propto \pi_{\text{old}}(\tau|\boldsymbol{x}) \exp(r(\boldsymbol{x}, \tau)/\beta)$, if we use an equivalent reward $r'(\boldsymbol{x}, \tau) = r(\boldsymbol{x}, \tau) + \zeta(\boldsymbol{x})$, the new un-normalized policy becomes:

$$\pi'_{\text{un-normalized}}(\tau|\boldsymbol{x}) \propto \pi_{\text{old}}(\tau|\boldsymbol{x}) \exp\left(\frac{r(\boldsymbol{x}, \tau) + \zeta(\boldsymbol{x})}{\beta}\right) = \left(\pi_{\text{old}}(\tau|\boldsymbol{x}) \exp\left(\frac{r(\boldsymbol{x}, \tau)}{\beta}\right)\right) \cdot e^{\zeta(\boldsymbol{x})/\beta}$$

When this expression is normalized over all trajectories $\tau$ to compute the final policy, the term $e^{\zeta(\boldsymbol{x})/\beta}$ is a constant factor that appears in both the numerator and the denominator (the partition function) and thus cancels out. This leaves the final policy unchanged. The similar logic applies to the PL model, where the scores for all trajectories are scaled by the same factor, resulting in an identical probability distribution over rankings.

**Practical Implications.** This invariance is highly valuable in practice. It provides the flexibility to shape the reward function to incorporate additional context without distorting the underlying learning problem of ranking trajectories. For example, in a production environment, a practitioner could define a cost-aware reward $r'(\boldsymbol{x}, \tau) = r(\boldsymbol{x}, \tau) - \text{cost}(\boldsymbol{x})$, where $r(\boldsymbol{x}, \tau)$ is the performance reward from Eq. 5 and $\text{cost}(\boldsymbol{x})$ is a penalty based on query complexity (Chen et al., 2023). The lemma guarantees that adding this query-dependent cost term $\zeta(\boldsymbol{x}) = -\text{cost}(\boldsymbol{x})$ does not change the optimal policy's preference for one demonstration sequence over another *for that given query*. Therefore, our approach of training $\pi_\theta$ to match $\pi^*$ remains a robust strategy focused on learning the optimal relative ordering of demonstration sequences.

