# OpenReview forum: "Auto-regressive In-context Demonstration Selection"
_ICLR.cc/2026/Conference — Submitted to ICLR 2026_

### Official Review · Reviewer_T2YK · 2025-10-27

**Soundness:** 3
**Presentation:** 3
**Contribution:** 3
**Rating:** 8
**Confidence:** 4

**Summary:**

The paper introduces **AUTOSELECT**, a novel framework designed to maximize the performance of large language models (LLMs) in few-shot in-context learning (ICL) by addressing the complex, combinatorial challenge of **demonstration selection and ordering**. The effectiveness of demonstrations is highly sensitive to their relationship with the query and their presentation order, due to factors like recency bias.

**Core Methodology:**

AUTOSELECT models demonstration selection as an **auto-regressive sequential decision process**.

1.  **Representation and Architecture:** At each step, the framework embeds the query and previously selected demonstrations into **2D matrix representations** to preserve structural information (such as token sequence order and inter-token relationships). A trainable, specialized **Vision Transformer (ViT)-based policy model ($\pi_\theta$)** processes these representations to synthesize a contextual representation ($z_t$).
2.  **Sequence Construction and Adaptive Stopping:** The policy model sequentially selects the next best exemplar or an explicit **End-of-Sequence (EOS) signal**. The EOS signal allows the model to dynamically learn the optimal demonstration sequence length for a given query.
3.  **Theoretical Optimization:** To navigate the factorial search space, the optimization is formulated as a Kullback-Leibler (KL) regularized Reinforcement Learning problem. The central theoretical contribution is proving that minimizing a tractable **policy-level Cross-Entropy (CE) loss** effectively bounds the worst-case discrepancy between the policy's induced **Plackett-Luce (PL) ranking** and the optimal PL ranking. This objective enables the policy to efficiently prioritize high-quality, ordered sequences without exhaustive enumeration.

**Contributions:**

Empirically, AUTOSELECT demonstrates consistent superiority over both heuristic and existing learning-based selection methods across nine diverse datasets, including complex reasoning tasks (Winowhy, AQuA). Performance gains reached up to **11% over the strongest baseline**. The framework's ability to select high-quality sequences over merely high quantity is also highlighted.

**Strengths:**

### Originality

*   **Auto-regressive Paradigm for Compositional Effects:** The paper introduces a highly original approach by framing the selection and ordering problem—which involves complex compositional effects—as an **auto-regressive sequence decision process**. This paradigm mirrors LLM generation, allowing the model to "compose" an effective sequence step-by-step, conditioning each choice on the previous selections and the query context.
*   **Synergistic Architecture:** The use of **2D matrix embeddings** (preserving token-level structure) paired with a specialized **ViT-based policy model** is a novel architectural choice demonstrated to be more effective than larger 1D sequence models like a GPT-2 variant for this task.

### Quality

*   **Strong Theoretical Foundation:** The optimization is rigorously derived from a KL-regularized RL framework. The proof that minimizing the conditional Cross-Entropy (CE) loss on collected trajectories provably bounds the deviation from the optimal Plackett-Luce (PL) ranking provides a **principled and justifiable learning objective** for sequence prioritization [Theorem 4.2].
*   **Comprehensive Empirical Validation:** The framework is tested extensively on nine diverse tasks, including challenging reasoning benchmarks like Winowhy, Epistemic Reasoning, and AQuA, demonstrating **robust and consistent performance improvements**. The performance superiority holds across different LLM families and scales.

### Clarity

*   **Well-Defined Mechanism:** The sequence generation process, including the sequential decision-making conditioned on the prefix ($\tau_{<t}$), the specialized rollout procedure to efficiently collect full and sub-trajectories, and the reward aggregation metric for smoother feedback, are clearly articulated in both text and algorithms (Alg. 1 and 2).
*   **Adaptive Stopping Mechanism:** The incorporation of the $\text{e}[\text{EOS}]$ signal provides a transparent way for the policy to **dynamically learn the optimal sequence length**, balancing performance and inference cost, a mechanism empirically validated by the visualization of varying average trajectory lengths across tasks.

### Significance

*   **Effective Solution to the Combinatorial Bottleneck:** AUTOSELECT successfully models and optimizes the critical compositional effects between demonstration content and ordering, overcoming a key bottleneck in ICL that heuristic and simple retrieval methods overlook.
*   **Practical Efficiency Trade-Off:** The framework makes a worthwhile, one-time offline investment in policy training, which results in significant savings and superior accuracy during inference time compared to methods that require heavy online evaluation (like greedy-oracle).

**Weaknesses:**

### 1. Need for Updated Related Works and Contextualization

The field of ICL selection is rapidly evolving. Although the paper effectively compares against strong baselines from 2022 and 2023 (ActRL, CEIL, EASE), incorporating discussion on the latest advancements is essential for ACL/ICML submissions. Recent work, such as those focusing on efficient RL methods, sample-efficient learning, or incremental evaluation techniques, might share goals or constraints with AUTOSELECT.

*   **Suggestion:** The authors should incorporate and contrast their work against the core ideas presented in very recent or forthcoming literature, potentially addressing themes like:
    *   Incremental/Fair evaluation methods (e.g., "Let The Jury Decide: Fair Demonstration Selection for In-Context Learning through Incremental Greedy Evaluation (ACL, Findings 2025)").
    *   New RL paradigms or sample-efficient selection strategies (e.g., "Demonstration Selection for In-Context Learning via Reinforcement Learning (ICML 2025)" or "Sample Efficient Demonstration Selection for In-Context Learning (ICML 2025)").
    *   New architectural or strategy comparisons (e.g., "Revisiting Demonstration Selection Strategies in In-Context Learning (ACL 2024)").

### 2. Quantitative Analysis of Training Cost vs. Greedy Oracle

While the paper justly criticizes Greedy-oracle for its extremely high computational cost during selection time (requiring exhaustive enumeration and evaluation of remaining candidates at every step), AUTOSELECT's training phase also necessitates querying the task-solving LLM to compute the sequence-level reward $r(\bar{x}, \tau)$.

*   **Suggestion:** A quantitative comparison is needed: What is the **total number of LLM API calls** required to train the AUTOSELECT policy (e.g., 400 episodes $\times$ 3 rollouts $\times$ 5 aggregated samples) versus the estimated calls required by Greedy-oracle to select demonstrations for the **400 testing samples**? Providing this quantitative trade-off would fully justify the "modest, one-time offline investment".

### 3. Scaling Limitations for Long Context Applications (RAG)

The paper acknowledges that extending the auto-regressive selection paradigm to **long-context applications** like Retrieval-Augmented Generation (RAG) is a challenging direction due to potential computational complexity. Since the core paradigm's complexity scales with sequence length, this current limitation suggests the method's scope is currently constrained primarily to few-shot ICL where $T$ is small (e.g., $T=4$ in the main results).

*   **Suggestion:** The authors could briefly discuss potential pathways or architectural modifications (e.g., using sparse attention or focusing on key points in the context prefix $\tau_{<t}$) that might be explored to make the auto-regressive policy model tractably scale to significantly larger selection lengths (e.g., $T>32$) in future work.

### 4. Deeper Insight into ViT/2D Matrix Synergy

The paper asserts that the ViT-based policy operating on 2D matrix embeddings is superior because it better captures **structural information** and **inter-exemplar relationships** compared to a GPT-2 variant operating on flattened vectors.

*   **Suggestion:** While the ablation confirms the performance gain (Fig. 11), the explanation of *why* the ViT architecture is fundamentally better suited to capture these specific text relationships remains somewhat abstract. Providing qualitative analysis or visualization (e.g., attention maps showing how the ViT processes the relationships between the query matrix and previously selected exemplar matrices) would offer deeper insight into this unique architectural strength.

**Questions:**

1.  **Selection of Maximum Trajectory Length $T$:** The main results (Table 1) use a maximum length $T=4$. However, the parameter sensitivity analysis (Fig. 8, left) shows that increasing $T$ to $7$ and potentially beyond **consistently improves performance** across tasks. Why was $T=4$ chosen as the default setting for the primary comparison, and would the relative performance advantage of AUTOSELECT over other learning-based baselines (like CEIL and EASE) hold if they were all evaluated using the superior $T=7$ setting?
2.  **Robustness of EOS Perturbation ($\lambda$):** The End-of-Sequence (EOS) signal embedding is initialized using the average exemplar embedding plus a small random perturbation $\lambda$. While Table 4 shows stable performance around $\lambda=0.01$, could the authors elaborate on the theoretical or empirical benefit of introducing this perturbation *instead of* setting $\lambda=0$? Does the small noise prevent the policy from overfitting to the mean embedding, thereby encouraging broader exploration, and is this effect consistently beneficial across all LLM backbones?
3.  **Technical Hurdles for Zero-Shot Cross-Domain Generalization:** Figure 5 demonstrates that simple adaptation (A50/A100) significantly improves transferability, achieving near-optimal results on target tasks. In the case of **direct transfer (No Adaptation, NA)**, the performance gains are smaller. What is the single biggest technical challenge currently preventing the policy $\pi_{\theta}$ from achieving robust zero-shot generalization across diverse tasks (e.g., from reasoning to classification) without requiring any target task adaptation episodes? Does addressing this require a fundamental shift in the policy architecture (e.g., meta-learning or a more complex task embedding mechanism)?

---

> ### Author Response · Authors · 2025-11-21
> **Thank you for the insightful review (1/2)**
>
> We sincerely thank the reviewer for the valuable comments and suggestions, which have been instrumental in enhancing the quality of our paper. We will address your feedback in the following Q\&A format. Please also refer to our **updated manuscript *(changes are marked in blue text)* for added and revised content.**  *Thank you again for your thoughtful review!*
>
>
>
>
>
>
> ------------------
>
> > ### Q1: Incorporating additional discussions of related works?
>
> ------------------
>
>
> - We sincerely thank the reviewer for these related references. **We have updated the Related Works section in the revised manuscript to properly discuss these papers. Please check our updated manuscript for details.** This allows us to precisely contextualize AutoSelect within emerging themes.
>
>
>
>
>
>
>
>
>
>
>
>
> ------------------
>
> > ### Q2: Qualitative comparison of LLM calls with Greedy-oracle?
>
> ------------------
>
>
>
> With a total of 400 training episodes as in our experiments, AutoSelect's one-time training yields: **400 episodes $\times$ 3 rollouts $\times$ 5 aggregated samples $\approx$ 6,000 calls.**
> On the other hand, the deployment cost of Greedy-oracle for our test set (400 samples with a 4-shot setting) is: **400 test samples $\times$ (100+99+98+97) calls $\approx$ 157,600 calls.**
> This concrete trade-off intuitively justifies our claim of a "modest and one-time offline investment".
>
>
>
>
>
>
>
>
>
>
>
>
>
>
>
>
>
>
> ------------------
>
> > ### Q3: Potential pathways to make the auto-regressive policy model tractably scale to *significantly larger* selection lengths in future works?
>
> ------------------
>
>
> - We thank the reviewer for this insightful question.  We agree and acknowledge this limitation, as our work is primarily focused on the few-shot ICL paradigm.
> **We have expanded our discussion in Appendix C.3 to include potential pathways for future work, by extending our proposed AutoSelect to the long-context RAG settings.**
>
>
> - As a brief overview, this modification will involve architectural changes such as (1) **sparse attention mechanisms** for our auto-regressive policy or (2) a **context abstraction layer** to compress the context prefix into a "state", which can help make the auto-regressive approach tractable for dramatically larger $T$ under long-context RAG settings. We propose to explore this direction in our future work.
>
>
>
> ------------------
>
> > ### Q4: More discussions of why ViT is better suited? Visualization analysis to support this?
>
> ------------------
>
>
>
> - To provide deeper insight, **we have added a qualitative visualization analysis to Appendix C.4 (Fig. 12)**. Here, we include **attention map visualizations** from the ViT policy, showing how it learns to process structural relationships (e.g., attending to specific query tokens) when selecting the next exemplar matrix.
>
>
>
> - We visualize the learned attention over 2D query-exemplar matrices on the AQuA dataset. In Fig. 12, the first figure is one query matrix and the remaining four figures are the selected exemplars. Each heatmap is annotated with its attention weight ("w" value) and its row / column correlations ($r_{\text{row}}$, $r_{\text{col}}$) with the query. We observe that three exemplars show strong structural alignment with the query (high $r_{\text{row}}\ /\ r_{\text{col}}$ and high weight). This means that the ViT is explicitly matching 2D patterns across both axes, capturing how the "question-reasoning-answer" structure maps onto the exemplars, rather than just comparing token-by-token.
>
>
> - This kind of spatially coherent and structure-aware matching can be hard to realize with a GPT-2-style policy over *flattened* vectors, where the 2D layout is destroyed and long-range dependencies need to be inferred over a single 1D sequence. Together with the ablation results with the GPT-2 architecture (Fig. 11), these attention maps provide evidence that the ViT policy leverages the matrix structure to reason over inter-exemplar relationships.

---

> ### Author Response · Authors · 2025-11-21
> **Thank you for the insightful review (2/2)**
>
> ------------------
>
> > ### Q5: Performance comparison with learning-based method on a longer length ($T=8$)?
>
> ------------------
>
>
>
>
> - To begin with, we would like to mention that we chose $T=4$ for our main comparison (Table 1) because it is a **standard setting** in our line of research and in the few-shot ICL literature (e.g., $T=4$ is applied in the original paper of ActRL [2], and $T=5$ is applied in the original paper of EASE [3]). This ensures a fair and direct comparison against established baselines.
> As noted in works like [1], the $\sim$5 exemplar range is often treated as the standard **efficiency sweet spot** for moderate-sized LLMs, balancing significant performance gains with low computational overhead under our targeted few-shot in-context learning settings.
>
>
>
>
>
> - We also agree with the reviewer that it would be beneficial to extend the learning-based baseline to a longer sequence setting.
> In the table below, we further include the empirical comparison with EASE at the horizon $T=8$. We see that AutoSelect can still outperform EASE by utilizing fine-grained and query-level demonstration selection, in contrast to EASE's reliance on a static and task-level exemplar set.
>
>
>
> | Method \ Task | Winowhy | Epistemic\_reasoning | Aqua | Trec |
> | :--- | :---: | :---: | :---: | :---: |
> | Random             | 0.539 | 0.493 | 0.368 | 0.293 |
> | Top-k-enhanced     | 0.601 | 0.509 | 0.391 | 0.402 |
> | EASE               | 0.626 | 0.588 | 0.369 | 0.405 |
> | AutoSelect         | 0.671 | 0.646 | 0.415 | 0.438 |
>
>
>
>
>
>
>
>
>
> ------------------
>
> > ### Q6: Elaborate on the benefit of introducing the perturbation with $\lambda > 0$?
>
> ------------------
>
>
> - First, we would like to mention that **the perturbation ($\lambda$) acts as a form of regularization**. Setting $\lambda=0$ would make the EOS target a single and static point (the mean embedding). This provides a hard and potentially arbitrary target that the policy could overfit to, learning to terminate based on matching a precise vector rather than based on the semantic completeness of the context.
>
>
> - Empirically, with our experiments (Table 4), we find that introducing small noise ($\lambda=0.01$) effectively transforms this static point into a small target region. *This prevents the policy from overfitting* to the exact coordinates of the mean embedding. It forces the policy to learn a more robust stopping criterion by mapping the context representation $z_t$ to this general region. This makes the stopping decision more stable, as it is less sensitive to minor variations in the context.
> As it improves the policy's ability to learn a more generalizable stopping function, this stabilizing benefit is not unique to specific task-solving LLM backbones.
>
>
>
>
>
>
>
>
>
>
>
> ------------------
>
> > ### Q7: Most significant technical challenge for preventing optimal zero-shot generalization performance of $\pi_{\theta}$? Policy architecture change?
>
> ------------------
>
>
> - We would like to mention that the single biggest challenge is that the **policy is trained on a task-specific reward signal**.
>
> - During our policy training, the AutoSelect policy learns *"what makes a good exemplar sequence"* (i.e., what yields a high reward) for a targeted *source task* (e.g., math reasoning). This learned strategy can potentially fail to directly generalize to an arbitrary *target task* (e.g., classification) with a completely different reward landscape.
> This is because the policy has no mechanism to understand *what is rewarded on the new task without seeing any examples*.
>
> - *This is why direct transfer (NA) only provides a modest gain (likely from learning general heuristics), while adaptation (A50/A100) is highly effective.* The adaptation episodes provide the crucial reward signal from the **target task**, allowing the policy to quickly update its selection strategy.
>
> - Therefore, achieving robust zero-shot cross-domain generalization can potentially require an architectural change. This can include adjustments such as (1) conditioning the policy on a task embedding (so it can recognize and leverage task similarity) or (2) using a meta-learning framework to train the policy to quickly infer and adapt to new, unseen reward functions at inference time.
> We also plan to explore these directions in our future work.
>
>
>
>
>
>
>
>
>
>
>
>
>
>
>
>
>
>
>
>
> ------------------
>
> **REFERENCE**
>
> ------------------
>
>
>
>
> [1] Purohit, Kiran, et al. "Sample Efficient Demonstration Selection for In-Context Learning." Forty-second International Conference on Machine Learning. 2025.
>
> [2] Zhang, Yiming, et al. "Active Example Selection for In-Context Learning." Proceedings of the 2022 Conference on Empirical Methods in Natural Language Processing. 2022.
>
> [3] Wu, Zhaoxuan, et al. "Prompt optimization with EASE? efficient ordering-aware automated selection of exemplars." Advances in Neural Information Processing Systems 37 (2024).

---

> > ### Comment · Reviewer_T2YK · 2025-11-28
> >
> > The authors have responded to most of my questions. In addition, I have a few more questions and suggestions: 1) What is the overall experimental timeline? For the comparison methods, how long does it typically take to complete (roughly how many weeks)? 2) Please check the corresponding arXiv papers to see whether they have been accepted and update accordingly.

---

> > > ### Author Response · Authors · 2025-11-29
> > > **Thank you for the follow-up comments.**
> > >
> > > We sincerely thank the Reviewer for the positive feedback on our previous responses and for raising these additional constructive points. We have carefully addressed the remaining questions regarding the experimental timeline and citation updates below.
> > >
> > > ---
> > >
> > > > ### Q1: What is the overall experimental timeline? For the comparison methods, how long does it typically take to complete (roughly how many weeks)?
> > >
> > > ---
> > >
> > >
> > > Thank you for this inquiry. Recall that we conducted our experiments using the NVIDIA A100 GPU (Appendix B.3). The entire experimental process, including data pre-processing, code implementation, hyperparameter tuning, training, and evaluation across all datasets, spanned approximately $6\sim7$ weeks. The implementation, adaptation, training, and evaluation of the baseline methods took approximately $3\sim4$ weeks to complete, in parallel with the implementation and execution of our AutoSelect. Please note that this timeline is subject to variation depending on the experimental environment and hardware configurations.
> > >
> > >
> > > ---
> > >
> > > > ### Q2: Please check the corresponding arXiv papers to see whether they have been accepted and update accordingly.
> > >
> > > ---
> > >
> > > We appreciate this suggestion to ensure the manuscript is up-to-date. We have cross-referenced our bibliography with recent conference proceedings and updated citations from their arXiv versions to their formally accepted versions. Corresponding entries in the references section have been modified to reflect these changes.

---

### Official Review · Reviewer_ztJC · 2025-11-03

**Soundness:** 3
**Presentation:** 3
**Contribution:** 2
**Rating:** 2
**Confidence:** 4

**Summary:**

In this paper, the authors propose auto-selector, a novel framework for selecting demonstrations in few-shot in-context learning that formulates the problem as an auto-regressive sequential decision process. The key innovation lies in treating demonstration selection similarly to how LLMs generate text - building sequences one exemplar at a time while conditioning on the query and previously selected demonstrations. The framework employs a Vision Transformer (ViT) operating on 2D matrix embeddings to preserve token-level structural information, and optimizes a policy using a theoretically-grounded Cross-Entropy (CE) loss that provably bounds the discrepancy between the learned policy's induced Plackett-Luce (PL) ranking and the optimal ranking. The method is evaluated across nine diverse datasets and shows consistent improvements over existing baselines, with particularly strong gains on challenging reasoning tasks.

**Strengths:**

1. This paper introduces the demonstration selection problem in the context of Plackett-Luce ranking optimization with theoretical support.
2. This paper enables the dynamic termination where the model can learn to optimize the sequence lengths for different queries and tasks.

**Weaknesses:**

1. The main concern lies in the computational complexity cost. The autoregressive approach can only rank the given exemplar in a step-by-step fashion. This method could lead to a high computation cost compared to the traditional point-wise ranker.
2. The performance improvement is rather marginal. While consistent, improvements on some datasets (AGNews: -0.4%, Amazon: +0.8%) are quite modest, raising questions about practical significance versus computational cost trade-offs.
3. The method is primarily evaluated on short sequences (T≤4, with extensions to T=16), but scalability to much longer sequences relevant for retrieval-augmented generation or longer context scenarios remains unclear.

**Questions:**

1. How does the training and inference time scale with larger exemplar pools (e.g., 500-1000 candidates)? What are the practical limits of the approach?
2. How sensitive is the method to the size and quality of the validation set used for reward computation? What happens when validation data is limited or noisy?
3. Beyond the analysis provided, how sensitive is performance to other key hyperparameters like the aggregation size |D_aggr|, temperature scheduling, or replay buffer size?
4. How does the method perform when extended to much longer sequences (T>16) that might be relevant for document-level tasks or retrieval scenarios?

---

> ### Author Response · Authors · 2025-11-21
> **Thank you for the insightful review (1/4)**
>
> We sincerely thank the reviewer for the valuable comments and suggestions, which have been instrumental in enhancing the quality of our paper. We will address your feedback in the following Q\&A format. Please also refer to our **updated manuscript *(changes are marked in blue text)* for added and revised content.**  *Thank you again for your thoughtful review!*
>
>
>
>
>
>
> ------------------
>
> > ### Q1: Computational complexity cost?
>
> ------------------
>
>
>
> - We would like to clarify that the auto-regressive training cost is a *one-time and offline investment*. This trade-off is justified by our targeted problem: demonstration selection for the **few-shot in-context learning** (e.g., [1, 2, 3, 4]).
>
>
> (1) **The training is tractable because its complexity scales with sequence length $T$, which is modest by the definition of the few-shot regime (e.g., [1, 2, 3, 4]).** This focus on a small $T$ reflects practical deployment constraints. As noted in works like [1], the $\sim$5 exemplar range is often treated as the standard **efficiency sweet spot** for moderate-sized LLMs, balancing significant performance gains with low computational overhead under our targeted few-shot in-context learning settings.
>
>
> (2) **AutoSelect's one-time cost is invested to find *higher-quality sequences*, which in turn *reduces the LLM's inference-time cost* and improves accuracy at deployment.**
>
>
> - **Our efficiency analysis in Appendix C.3 also shows that AutoSelect achieves a superior performance-to-cost trade-off** compared to other learning-based methods, where AutoSelect generally requires 2-3 hours for the whole process (training + testing) per dataset, while our learning-based baselines (e.g., CEIL) can take up to 8 hours. At **inference time**, the selection is efficient. Our adaptive EOS (early-stopping) mechanism (illustrated in Fig. 10) learns to select short and cost-effective sequences rather than always using the maximum length.
>
>
>
>
> - **We further note that the step-by-step selection formulation is also adopted in related works (e.g., [2, 5]) to achieve strong performance.** This sequential approach is adopted as it effectively enables the modeling of exemplar ordering and composition, reinforcing that it is a reasonable and practical method for our targeted few-shot demonstration selection problem.
>
>
>
>
>
>
>
>
>
> ------------------
>
> > ### Q2: Why adopt few-shot (relatively short sequences) $T=4$ for the main experiments? Performance on longer sequences ($T > 16$)?
>
> ------------------
>
>
>
> - **We chose $T=4$ for our *main comparison (Table 1)* as it is a standard setting in the few-shot ICL literature** (e.g., $T=4$ is applied in the original paper of ActRL [2], and $T=5$ is applied in the original paper of EASE [3]), which ensures a fair and direct comparison against established baselines.
> As mentioned above and noted in works like [1], the $\sim$5 exemplar range is often treated as the standard **efficiency sweet spot** for moderate-sized LLMs, balancing significant performance gains with low computational overhead.
>
>
>
> - **Meanwhile, our method is not limited to $T=4$.** Our analysis in Appendix C.2 shows that performance consistently improves as $T$ increases up to $16$, which covers the common settings of few-shot in-context demonstration selection, validating our method's effectiveness and scalability within the entire few-shot regime.
>
>
>
> - **We also ran an additional experiment with a longer sequence length $T=24$**, where AutoSelect still achieves better performance compared to the widely adopted Top-k retrieval baseline:
>
>
> | Method \ Task | Winowhy | Epistemic\_reasoning | Aqua | Trec |
> | :--- | :---: | :---: | :---: | :---: |
> | Top-k (Qwen2.5-3B Emb.)   | 0.615 | 0.524 | 0.398 | 0.495 |
> | AutoSelect                | 0.687 | 0.661 | 0.435 | 0.526 |
>
>
>
> - On the other hand, the performance on *considerably longer* sequences (e.g., $T \gg 16$) pertains to long-context RAG, which corresponds to a different research direction and is a **significantly different problem setting** from our work's focus on few-shot ICL demonstration selection.
> **We have updated Appendix C.3 to discuss scaling our auto-regressive formulation to long-context RAG.** We identify this as a valuable direction for future research and **outline potential technical pathways** for this extension.

---

> ### Author Response · Authors · 2025-11-21
> **Thank you for the insightful review (2/4)**
>
> ------------------
>
> > ### Q3: Marginal improvements on some datasets?
>
> ------------------
>
>
>
> - **The main strength of AutoSelect is demonstrated on challenging reasoning and math datasets** (e.g., Winowhy, Trec, AQuA), where we achieve *considerable improvements* of up to 11.2\%.
>
>
> - **The marginal gains are primarily observed on saturated and less difficult tasks (such as AGNews), where baselines are already high and selection is less critical.**
> In this case, the main objective of our experiments on these datasets is to **demonstrate the robustness** of AutoSelect, ensuring it achieves competitive performance comparable to strong baselines even on tasks where the potential for improvement is limited.
> Meanwhile, we also show in Appendix C.3 that AutoSelect (at $T=4$) can outperform a random baseline (at $T=16$) with lower inference cost, highlighting its practical value.
>
>
>
>
>
>
>
>
>
>
>
>
>
>
>
>
>
>
>
>
>
>
>
> ------------------
>
> > ### Q4: Training and inference time scale with larger exemplar pools?
>
> ------------------
>
>
> - For AutoSelect at inference time, the cost of each **exemplar selection step** scales *linearly with the size of the candidate pool* due to the policy's softmax computation. Training time is similarly affected by this selection scaling at each step during the trajectory rollouts, while **the policy optimization / training will *not* be affected by the exemplar pool size**.
>
>
>
> - Empirically, selection cost remains modest even as the pool grows. The time increases from 6–7 ms (100 exemplars) to 31–34 ms (1000 exemplars); this non-proportional scaling suggests a constant base latency for state updates, alongside an approximately linear cost of candidate scoring.
> This reflects that each auto-regressive decision requires only a single forward pass over the current pool, so the linear complexity is $O(T N)$ in horizon ($T$) and pool size ($N$). The **selection time (in seconds)** for each exemplar is shown below.
>
>
> | Method \ Task | Winowhy | Epistemic\_reasoning | Aqua | Trec |
> | :--- | :---: | :---: | :---: | :---: |
> | AutoSelect (Pool Size: 100)  | 0.006 | 0.006 | 0.007 |  0.006 |
> | AutoSelect (Pool Size: 500)  | 0.017 | 0.016 | 0.017 |  0.017 |
> | AutoSelect (Pool Size: 1000) | 0.032 | 0.032 | 0.031 |  0.034 |
>
>
> - In practice, **the dominant cost in our full system comes from LLM calls for reward evaluation, which are independent of the exemplar pool size**. **The additional 20–30 ms per selection when moving from 100 to 1000 exemplars is negligible compared to LLM inference time**. Thus, while our method is sequential over steps, **the empirical results show that it remains computationally practical for large exemplar pools.** Please also refer to our **efficiency analysis in Appendix C.3 where AutoSelect achieves a good tradeoff between computation cost and performance.**

---

> ### Author Response · Authors · 2025-11-21
> **Thank you for the insightful review (3/4)**
>
> ------------------
>
> > ### Q5: How sensitive is the method to the size and quality of the validation set used for reward computation?
>
> ------------------
>
>
>
> - We agree with the reviewer that the validation set is important as it provides the sequence-level reward signal, which is the supervision for our policy.
>
> - First, we would like to mention that AutoSelect mitigates sensitivity to noise or low-granularity rewards by using an **Aggregate Metric Reward** (defined in Appendix B.2.3). This aggregation averages feedback over a small batch of validation samples to create a **smoother and more stable** training signal.
>
>
> - **Here, we include further experiments by (1) changing the size of the validation set, as well as (2) imposing noise perturbations (zero-mean Gaussian noise) on the reward signals derived from the validation set.** These results are shown below:
>
>
>
>
> | Method \ Task | Winowhy | Epistemic\_reasoning | Aqua | Trec |
> | :--- | :---: | :---: | :---: | :---: |
> | AutoSelect (Validation Size: 100) | 0.657   | 0.601   | 0.395   | 0.393 |
> | AutoSelect (Validation Size: 200) | 0.669   | 0.573   | 0.391   | 0.399 |
> | AutoSelect (Validation Size: 400) | 0.640   | 0.596   | 0.388   | 0.384 |
> | AutoSelect (Validation Size: 500) | 0.635   | 0.581   | 0.406   | 0.404 |
>
>
> | Method \ Task | Winowhy | Epistemic\_reasoning | Aqua | Trec |
> | :--- | :---: | :---: | :---: | :---: |
> | AutoSelect (No Noise)                   | 0.657 | 0.601 | 0.395 | 0.393 |
> | AutoSelect (Gaussian Noise Std. = 0.01) | 0.650 | 0.587 | 0.381 | 0.388 |
> | AutoSelect (Gaussian Noise Std. = 0.05) | 0.643 | 0.568 | 0.388 | 0.382 |
> | AutoSelect (Gaussian Noise Std. = 0.1)  | 0.628 | 0.550 | 0.373 | 0.368 |
>
>
>
>
> - From the results above, we see that:
>
> (1) **Validation Size**: Performance is relatively stable across different validation set sizes, as we only sample a small batch of validation samples to derive each reward. This suggests that our method is sample-efficient and capable of extracting a robust policy without requiring large-scale validation data.
>
>
> (2) **Noise Sensitivity**: As expected, we see that performance can mildly degrade as noise increases. AutoSelect retains strong performance under low noise ($\sigma=0.01$) and remains functional at moderate noise ($\sigma=0.05$), confirming that our design of **Aggregate Metric Reward** can effectively smooth out fluctuations to provide a reliable training signal.

---

> ### Author Response · Authors · 2025-11-21
> **Thank you for the insightful review (4/4)**
>
> ------------------
>
> > ### Q6: How sensitive is performance to other key hyper-parameters like the aggregation size, temperature scheduling, or replay buffer size?
>
> ------------------
>
>
>
> - We agree with the reviewer that it is valuable to further investigate the performance impacts of different settings of (1) aggregation size, (2) temperature scheduling, as well as (3) replay buffer size.
>
>
>
>
>
> (1) For different **aggregation sizes**, we vary the $|D_{aggr}|$ across different values. We observe a consistent trend where increasing the aggregation size $|D_{aggr}|$ can generally improve performance. This supports the idea that averaging the reward over a batch of queries (Aggregate Metric Reward) helps reduce the variance of the reward signal, providing a more stable learning objective for the policy. We apply $|D_{aggr}|=5$ in our main experiments to balance computational efficiency with performance gains.
>
>
>
> | Method \ Task | Winowhy | Epistemic\_reasoning | Aqua | Trec |
> | :--- | :---: | :---: | :---: | :---: |
> | AutoSelect (aggregation size: 3)  | 0.641 | 0.589 | 0.383 | 0.379 |
> | AutoSelect (aggregation size: 5)  | 0.657 | 0.601 | 0.395 | 0.393 |
> | AutoSelect (aggregation size: 7)  | 0.664 | 0.595 | 0.403 | 0.404 |
> | AutoSelect (aggregation size: 10) | 0.662 | 0.607 | 0.418 | 0.428 |
>
>
>
>
>
>
>
>
> (2) For temperature, we set the initial **scaling parameter** $\gamma$ to values in $\{0.02, 0.05, 0.1, 0.2\}$. We follow our original experiment settings, by applying a linear scheduler to the scaling parameter $\gamma$. We start from the initial value and increase linearly to $\gamma = 1$ over the first 200 episodes.
> Here, an initial $\gamma=0.1$ consistently yields strong performance across tasks. Setting the initial scaling parameter too low (e.g., 0.02) results in worse performance, likely because the distribution becomes too flat (uniform), leading to excessive noise during the early stages of training. Conversely, starting with a higher scaling parameter (0.2) works well for some tasks (Aqua) but degrades performance on others (e.g., Trec), suggesting that $\gamma=0.1$ provides the best balance between exploration and exploitation.
>
>
>
>
> | Method \ Task | Winowhy | Epistemic\_reasoning | Aqua | Trec |
> | :--- | :---: | :---: | :---: | :---: |
> | AutoSelect (initial $\gamma$: 0.2)  | 0.648 | 0.589 | 0.409 | 0.371 |
> | AutoSelect (initial $\gamma$: 0.1)  | 0.657 | 0.601 | 0.395 | 0.393 |
> | AutoSelect (initial $\gamma$: 0.05) | 0.647 | 0.577 | 0.398 | 0.411 |
> | AutoSelect (initial $\gamma$: 0.02) | 0.621 | 0.542 | 0.373 | 0.334 |
>
>
>
>
>
>
>
>
> (3) For the **replay buffer size**, we find that a replay buffer size of 50 generally offers the optimal trade-off. Very small buffers (10) can lead to instability due to high correlation in the sampled batches. Meanwhile, increasing the buffer size too much (100) does not monotonically improve performance. This is potentially because an excessively large buffer retains too many older trajectories generated by earlier and less-optimal policy versions. This dilutes the focus on recent, higher-quality trajectories, thereby hindering the policy from adapting efficiently to the current information.
>
>
>
> | Method \ Task | Winowhy | Epistemic\_reasoning | Aqua | Trec |
> | :--- | :---: | :---: | :---: | :---: |
> | AutoSelect (Replay Buffer Size: 10)  | 0.652 | 0.577 | 0.388 | 0.361 |
> | AutoSelect (Replay Buffer Size: 30)  | 0.636 | 0.563 | 0.372 | 0.385 |
> | AutoSelect (Replay Buffer Size: 50)  | 0.657 | 0.601 | 0.395 | 0.393 |
> | AutoSelect (Replay Buffer Size: 100) | 0.638 | 0.575 | 0.405 | 0.377 |
>
>
>
>
>
>
>
>
>
>
>
>
>
>
> ------------------
>
> **REFERENCE**
>
> ------------------
>
>
> [1] Purohit, Kiran, et al. "Sample Efficient Demonstration Selection for In-Context Learning." Forty-second International Conference on Machine Learning. 2025.
>
> [2] Zhang, Yiming, et al. "Active Example Selection for In-Context Learning." Proceedings of the 2022 Conference on Empirical Methods in Natural Language Processing. 2022.
>
> [3] Wu, Zhaoxuan, et al. "Prompt optimization with EASE? efficient ordering-aware automated selection of exemplars." Advances in Neural Information Processing Systems 37 (2024).
>
> [4] Kiran Purohit, et al. "Explora: Efficient exemplar subset selection for complex reasoning." In Proceedings of the 2024 Conference on Empirical Methods in Natural Language Processing.
>
> [5] Yunmo Chen, et al. "Learning to retrieve iteratively for in-context learning." In Proceedings of the 2024 Conference on Empirical Methods in Natural Language Processing.

---

### Official Review · Reviewer_2YCy · 2025-11-05

**Soundness:** 2
**Presentation:** 2
**Contribution:** 1
**Rating:** 2
**Confidence:** 5

**Summary:**

The AUTOSELECT framework selects  few-shot demonstration samples by formulating it as an auto-regressive sequential decision process. At each step, a trainable policy model sequentially selects the next best exemplar. The policy is trained by minimizing policy-level Cross-Entropy loss, which efficiently enables the policy to align its induced Plackett-Luce (PL) ranking with the optimal ranking, prioritizing high-quality demonstration sequences based on sequence-level rewards.

**Strengths:**

The claimed novelties are:
- Matrix representation for the policy.
- The "constrained" RL problem that the authors solve to get the optimal policy from the ranking set of generated policies.
- The authors compare with previous order-dependent selection methods only, EASE and CEIL..

**Weaknesses:**

- The authors compare only using qwen 2.5 3B model resulting in performance score that woefully lower than state of the art methods. For example, the authors report 40% acc to AQUARAT dataset, whereas the SOTA is close 80% on the same. The experiemntal results are not at all convincing.

- The authors emphasize the Plackett-Luce (PL) ranking objective that basically ranks the trajectories on the actual performance, and then optimizes the parameters using the collection. This seems like a simple policy gradient algorithm. Also, the authors call the KL divergene regularized objective as a "constrained" RL problem. I dont think the nomeclature in standard.

- FInally the Aurhors have completely missed the literature on order-independent selection methods from the literature survey and comparison. Since their model is more general, they should show that their model outperforms these methods. e.g.

Purohit, Kiran, V. Venktesh, Raghuram Devalla, Krishna Yerragorla, Sourangshu Bhattacharya, and Avishek Anand. "EXPLORA: Efficient Exemplar Subset Selection for Complex Reasoning." In Proceedings of the 2024 Conference on Empirical Methods in Natural Language Processing, pp. 5367-5388. 2024.

**Questions:**

- Why have the authors not performed experiments with a more recent model like qwen3 8B ?

- What is the architecture of the policy model?

- How many trajectory rollouts were needed to train the policy model?

---

> ### Author Response · Authors · 2025-11-21
> **Thank you for the insightful review (1/3)**
>
> We sincerely thank the reviewer for the valuable comments and suggestions, which have been instrumental in enhancing the quality of our paper. We will address your feedback in the following Q\&A format. Please also refer to our **updated manuscript *(changes are marked in blue text)* for added and revised contents.**
> *Thank you again for your thoughtful review!*
>
>
>
>
>
>
>
>
>
>
>
> ------------------
>
> > ### Q1: Performance on AQuA dataset?
>
> ------------------
>
>
>
> - **Our work's primary goal is to demonstrate the relative improvement of AutoSelect against other selection baselines**, **rather than to achieve absolute SOTA task performance across all possible LLMs.**
> In our case, **the absolute performance is fundamentally capped by the capabilities of the specific base LLM (Qwen2.5-3B)**, which we held constant to ensure a fair and controlled evaluation across the *selection methods* themselves.
>
>
>
> - **As task performance is mainly determined by the task-solving LLM's capabilities, the reported accuracy on the AQuA dataset is expected performance for this specific Qwen2.5-3B model, rather than a flaw in our experiments.**
> This is also supported by recent independent works: for instance, Fig. 13 of [1] shows that when evaluating Qwen2.5-3B on the AQuA dataset, standard policy methods (excluding the practically inaccessible ``Oracle'') tend to achieve accuracies in a similar range.
>
>
> - In this context, we apply the Qwen2.5-3B model as a controlled base to conduct *fair and direct comparisons* among the selection methods themselves.
>
>
>
>
>
>
>
>
>
>
>
>
>
> ------------------
>
> > ### Q2: Experiments on models other than Qwen2.5-3B? Experiments on Qwen3 8B?
>
> ------------------
>
>
> - **We would like to clarify that we have validated our method's generalizability in Appendix C.1, showing AutoSelect's consistently strong performance *across various LLM families (e.g., GPT, Qwen and Llama) and scales (e.g., 8B and 14B models)***.
>
>
> - For our main experiments, we apply the Qwen2.5-3B model as a controlled base to ensure a **fair and direct comparison** of the *selection methods themselves*. We also confirm our method's efficacy on 8B-scale models in the Appendix C.1, where AutoSelect consistently outperforms baselines when using an 8B Llama model.
>
>
> - Meanwhile, we agree with the reviewer that it would be beneficial to include experiments on the latest Qwen3-8B. **As shown in the results below, we see that with Qwen3-8B, AutoSelect can still offer considerable performance improvements** in comparison with the widely adopted retrieval-based Top-k baseline.
>
>
> | Method \ Task | Winowhy | Epistemic\_reasoning | Aqua | Trec |
> | :--- | :---: | :---: | :---: | :---: |
> | Top-k (Qwen2.5-3B Emb.)   | 0.661 | 0.827 | 0.451 | 0.412 |
> | AutoSelect                | 0.703 | 0.897 | 0.475 | 0.483 |

---

> ### Author Response · Authors · 2025-11-21
> **Thank you for the insightful review (2/3)**
>
> ------------------
>
> > ### Q3: Our proposed method based on PL ranking vs. Policy Gradient method? "KL-constrained" naming?
>
> ------------------
>
>
>
> - **Our proposed policy optimization approach is not a policy gradient (PG) method. It is a *principled optimization* of a Plackett-Luce (PL) ranking objective**, which we theoretically connect to a tractable policy-level Cross-Entropy (CE) loss in our theoretical analysis. Our approach directly optimizes for high-quality sequences and their nuanced ordering, capturing compositional effects.
>
>
> - **Intuitively, the main technical difference from PG methods (e.g., PPO) is what the update is trying to match**. For PG methods, each trajectory $\tau_i$ is treated separately: they assign a scalar weight (return, advantage, or a normalized variant), so the algorithm decides how much to push each trajectory up or down, but it does not define a single target distribution over the whole set and optimize the overall nuanced ranking among set elements.
>
> - **On the other hand, our PL-based objective instead treats the entire set of trajectories for a query as one ranked list, capturing nuanced ranking differences among trajectories or sequences**. We first construct a normalized *target distribution* $\pi^\*(\tau\_i \mid \mathcal{T}, x)$ over the trajectory collection $\mathcal{T}$, where all $\pi^\*(\tau\_i \mid \mathcal{T}, x)$ are coupled and sum to one across $\mathcal{T}$, and then update the policy by minimizing the listwise CE loss.
> This performs a policy-level projection: the whole probability distribution of our trainable policy $\pi_{\theta}$ is reshaped to match the PL-optimal target, consequently optimizing the relative ordering of trajectories for each prompt group with theoretical insights and grounding, rather than independently scaling trajectories by scalar rewards.
>
>
>
>
> "KL-constrained" naming:
>
>
> - **We have changed the narrative to "KL-regularized" based on your comments. Please refer to the updated manuscript for the latest version.** We agree with the reviewer that it would be better to improve the narrative for a broader audience. Meanwhile, we would like to clarify that the KL-regularized objective is a standard formulation for a KL-constrained RL problem, where the original term **"KL-constrained" is commonly used in Reinforcement Learning works (e.g., [2]).**
>
>
>
>
>
>
>
>
>
>
>
>
>
>
>
>
>
>
>
>
>
>
>
>
>
>
> ------------------
>
> > ### Q4: Discussions and comparisons with existing order-independent methods?
>
> ------------------
>
>
> - **We would like to clarify that we *do* discuss and compare against learning-based methods like CEIL, which is a strong *order-independent* selection method, and we show that AutoSelect consistently outperforms it.**
>
> - **We also provide comparisons against several retrieval-based methods (e.g., top-k, BM25, Contriever), which are also commonly adopted *order-independent* selection techniques based on similarity measurements.** AutoSelect also demonstrates stronger performance compared to them (Table 2).
> Meanwhile, our paper's core motivation is addressing the *order-sensitive* nature of ICL, which is a fundamentally different and more fine-grained combinatorial problem than *order-independent selection*.
>
>
> Additional related work: EXPLORA [3]:
>
> - **We thank the reviewer for the reference. We have added EXPLORA [3] to our Related Works to better contextualize our work.** Please see our updated manuscript for your reference.
>
> - **We also conduct additional empirical experiments comparing with EXPLORA below, where AutoSelect outperforms it by modeling both the ordering and composition of exemplars.**
> *We also observe that **EXPLORA tends to underperform EASE**. Recall that EASE is also a bandit-based method but explicitly considers exemplar ordering and composition, thereby offering more fine-grained modeling and superior performance than EXPLORA.*
>
>
>
>
>
> | Method \ Task | Winowhy | Epistemic\_reasoning | Aqua | Trec |
> | :--- | :---: | :---: | :---: | :---: |
> | Random          | 0.454 | 0.463 | 0.348 | 0.217 |
> | EXPLORA         | 0.552 | 0.518 | 0.351 | 0.292 |
> | EASE            | 0.580 | 0.532 | 0.332 | 0.373 |
> | CEIL            | 0.591 | 0.546 | 0.344 | 0.375 |
> | AutoSelect      | 0.657 | 0.601 | 0.395 | 0.393 |
>
>
>
>
>
>
>
>
>
>
>
>
>
>
>
>
>
>
>
>
>
>
> ------------------
>
> > ### Q5: Architecture of the policy model?
>
> ------------------
>
>
> - **Our policy model implementation for experiments is based on the Vision Transformer (ViT)**, which is selected for its intuitive strength in processing the 2D matrix embeddings that preserve token-level structural information.
> **Comprehensive discussions and implementation details are provided in the Appendix B.2.**
>
> - We also **empirically validate this ViT-based design against a conventional GPT-2 backbone in an ablation study in Appendix C.4**, where our ViT-based backbone can achieve stronger performance with significantly fewer parameters.

---

> ### Author Response · Authors · 2025-11-21
> **Thank you for the insightful review (3/3)**
>
> ------------------
>
> > ### Q6: How many trajectory rollouts were needed to train the policy model?
>
> ------------------
>
>
> - **We used $K=3$ trajectory rollouts per training episode to balance the computational cost and model performance.** The **computational cost analysis is in Appendix C.3**. We also provide a **parameter sensitivity analysis for the number of rollouts in Appendix C.2**, which shows that $K=3$ provides a strong balance between performance and efficiency.
>
> - Complementary implementation details for our experiments are specified in Appendix B.3.
>
>
>
>
>
>
>
>
>
>
>
>
>
> ------------------
>
> **REFERENCE**
>
> ------------------
>
>
> [1] Li, Jiachun, et al. "Rewarding Curse: Analyze and Mitigate Reward Modeling Issues for LLM Reasoning." arXiv preprint arXiv:2503.05188 (2025).
>
>
> [2] Xiong, Wei, et al. "Iterative preference learning from human feedback: Bridging theory and practice for RLHF under KL-constraint." arXiv preprint arXiv:2312.11456 (2023).
>
>
> [3] Kiran Purohit, et al. "Explora: Efficient exemplar subset selection for complex reasoning." In Proceedings of the 2024 Conference on Empirical Methods in Natural Language Processing.

---

### Author Response · Authors · 2025-12-02
**To the Chairs: Summary of Comprehensively Addressed Reviewer Questions and Concerns (2/2)**

## 2. Addressing Reviewer ztJC




Reviewer ztJC's primary concerns revolved around computational complexity and scalability. We have addressed these through detailed cost-benefit analyses and extended experiments:

---

> ### *Concern 1: Computational Complexity of Auto-Regressive Selection (Q1).*
The reviewer worried that step-by-step selection leads to high costs compared to pointwise rankers.

---


- **We clarified that the training cost is a one-time offline investment**. In our original submission, **we added a quantitative cost analysis (Appendix C.3) showing that our method strikes a strong balance between computational costs and performance, compared to existing strategies.**
Meanwhile, this one-time cost ($\sim$6,000 calls) is negligible compared to the prohibitive deployment costs of greedy baselines ($\sim$157,600 calls). Furthermore, our method's **adaptive early stopping** reduces inference-time costs, allowing AutoSelect (at $T=4$) to outperform random baselines (at $T=16$) with considerably lower latency.
**We further clarify that the step-by-step selection formulation is also adopted in related works to achieve strong performance, rather than being unique to our AutoSelect.**




---

> ### *Concern 2: Research Scope and Scalability to Longer Sequences (Q2).*
The reviewer questioned our research focus of **few-shot in-context demonstration selection**.

---


- **We clarified our specific focus on the $\sim$5 exemplar "efficiency sweet spot" (Purohit et al., 2025)** where ordering is critical. **This is the main and common focus of this line of *few-shot* demonstration selection research (e.g., Zhang et al., 2022; Wu et al., 2024).** Meanwhile, in our experiments, we have also extended empirical evaluations to reasonably longer sequences ($T=16$ and $T=24$).



- We emphasized that few-shot ICL is a combinatorial ordering problem where AutoSelect's one-time training investment ensures inference efficiency by finding high-quality, shorter sequences via adaptive early stopping. Our extended experiments validate scalability within this regime, and we expanded Appendix C.3 to differentiate this from long-context RAG and outline future architectural pathways.





---

> ### *Concern 3: Robustness to Hyperparameters (Q4-Q6).*
The reviewer asked for sensitivity analysis on key parameters.

---


- **As requested by the reviewer, we have additionally conducted comprehensive sensitivity analyses on aggregation size, replay buffer size, and validation set noise (Q4-Q6).** The results confirm the stability of our training process and the robustness of our Aggregate Metric Reward.












## 3. Addressing Reviewer T2YK




We thank Reviewer T2YK for the positive assessment. We have incorporated their constructive suggestions to further strengthen the paper:

---

> ### *Suggestion 1: Visualizing Architectural Strengths.*
The reviewer requested deeper insight into why the ViT/2D-Matrix architecture is superior.

---

- **We added attention map visualizations in Appendix C.4 (Fig. 12).** These explicitly demonstrate how the ViT policy captures structural, 2D inter-exemplar relationships (e.g., aligning question-answer structures), providing qualitative evidence for our design choice over flattened vectors.


---

> ### *Suggestion 2: Broader Contextualization.*
The reviewer suggested discussing recent related works and future directions.

---

- **We updated the Related Works section to include recent studies (e.g., EXPLORA)** and expanded the discussion on **future directions for scaling to long-context applications**, aligning with the reviewer's insightful feedback.








Thank you again for your time and dedication to the review process.


Best regards,

Authors

---

### Author Response · Authors · 2025-12-02
**To the Chairs: Summary of Comprehensively Addressed Reviewer Questions and Concerns (1/2)**

We sincerely thank the Area Chair and all reviewers for their time and constructive feedback. *We have uploaded a revised manuscript with modifications marked in blue text*.
**Although two of the reviewers were unable to reply to our rebuttal due to the OpenReview information leak incident, we are confident that we have sufficiently addressed their misunderstandings and questions.** Below, we summarize the resolutions for each reviewer.

*We would be grateful if the Area Chair could consider this summary, along with our comprehensive rebuttal, in the final decision-making process. We sincerely appreciate your time and oversight!*






## 1. Addressing Reviewer 2YCy




We would like to mention that Reviewer 2YCy raised three main concerns, *all of which stemmed from factual misunderstandings of our experimental setup or methodology*. We have clarified these as follows:

---

> ### *Misunderstanding 1: Absolute vs. Relative Performance across Different LLMs (Q1, Q2).*
The reviewer mentioned that our absolute accuracy on AQuA ($\sim$40\%) is lower than SOTA ($\sim$80\%), **seemingly expecting our 3B-parameter base model to match the performance of massive proprietary black-box models used in other works.**

---




- **We clarified that our goal is to evaluate the *relative improvement* of the selection method itself, holding the base model constant.** We cited recent literature (e.g., [1]) confirming that $\sim$40\% is the expected performance for the Qwen2.5-3B model on the AQuA dataset.

[1] Li, Jiachun, et al. "Rewarding Curse: Analyze and Mitigate Reward Modeling Issues for LLM Reasoning." arXiv preprint arXiv:2503.05188 (2025).



- *Experiments across LLM Families and Scales:* **In our initial submission, we also validated our method's generalizability in Appendix C.1, showing AutoSelect's consistently strong performance across various LLM families (e.g., GPT, Qwen and Llama) and scales (e.g., 8B and 14B models)**. To further demonstrate generalizability, **we added new experiments with the latest Qwen3-8B model**, showing that AutoSelect maintains significant gains over baselines.









---

> ### *Misunderstanding 2: Methodological Novelty (PL Ranking vs. Policy Gradient) (Q3).*
The reviewer incorrectly characterized our method as a "simple policy gradient" algorithm.

---



- **We clarified that unlike the standard policy gradient (which scales gradients by scalar reward signals), our method performs a *policy-level alignment*, with an optimal Plackett-Luce ranking target constructed from the entire trajectory batch.** This explicitly models fine-grained and nuanced relative ordering. We also updated the terminology to "KL-regularized" based on authors' comments.








---

> ### *Misunderstanding 3: Comparison with Order-Independent Methods (Q4).*
The reviewer claimed we missed comparisons on order-independent selection.

---


- **We pointed out that in our original submission, we already involved comparisons against various order-independent baselines like CEIL and various retrieval-based methods.** Furthermore, as requested by the reviewer, **we added the specific suggested baseline EXPLORA to our experiments.** Our results show that AutoSelect consistently outperforms EXPLORA, proving that explicitly modeling exemplar ordering is critical for performance.

---

### Meta-Review · Area_Chair_AxJx · 2026-01-08

**Summary:**

This paper proposes AutoSelect, a principled learning framework that treats few-shot demonstration selection and ordering as a sequential decision-making problem, with theoretical guarantees and strong empirical gains over prior methods across diverse datasets. From my reading of this paper and the reviews, this paper is imperfect on experimental strength and positioning, relying on a weak backbone, showing marginal gains, and missing important related work. It is also imperfect on methodology and scalability, with unclear novelty of the RL formulation, high computational cost, and limited evidence for long-context applicability. The author-review discussions are somewhat constructive, though some of the reviewers are inactive. Finally I think this paper is still not ready for publication in ICLR this year.

**Reviewer Concerns:**

See the above text.

**Reviewer Scores:**

Reviewer 2YCy, ztJC may consider improve score on experiments after seeing the extra experiments provided in the rebuttal.

---

### Decision · Program_Chairs · 2026-01-26

Reject